# Stable Planning through Aligned Representations in Model-Based Reinforcement Learning

## Abstract

Integrating planning with reinforcement learning (RL) significantly improves problem-solving capabilities for sequential decision-making problems, particularly in sparse-reward, long-horizon tasks. Recently, it has been shown that discrete world models can be trained such that no model degradation occurs over thousands of time steps and states can be re-identified during planning. As a result, a heuristic function can be trained with data generated from the world model, and the learned world model and heuristic function can be used with planning to solve problems. However, this approach fails to solve problems with state transformations to which the world model and heuristic function should be invariant (i.e., noise), without re-training the world model and heuristic function. In this work, we introduce Stable Planning through Aligned Representations (SPAR), an efficient framework that trains a discrete world model and heuristic function in a clean Markov decision process (MDP) and trains an alignment network to map transformed states to their discrete latent state in the clean MDP. When solving problems, we exploit the underlying discrete latent representation and round the output of the alignment network with the aim of exactly matching the clean latent state. As a result, adapting to transformations only requires training the adaptation network while the world model and heuristic function remain fixed. We then demonstrate its effectiveness on Rubik's Cube and Sokoban domains, and compare it with applying a similar approach to a world model with continuous latent representations. SPAR successfully solves over 90% of problems with 17 different visual transformations and real-world images. This adaptation process requires no additional world model or heuristic function re-training, and reduces re-training time by at least 95%.

## 1 Introduction

Many real-world problems require long-horizon planning to find solutions. However, planning cannot happen when the state transition function (also known as the "world model"), which maps states and actions to next states, is not known. One approach used to address this, which has yielded success in both simulation and the real-world, is to learn the world model using data obtained from interacting with the environment (Racanière et al., 2017; Hafner et al., 2019a; Bagatella et al., 2021; Tian et al., 2021; Agostinelli & Soltani, 2024). These methods then use the learned world model to train a heuristic function, which either estimates the expected future reward, in the general reinforcement leanring setting, or remaining cost to go to the goal (also known as the "cost-to-go"), in the more restricted pathfinding setting. Given a test instance, the learned world model and heuristic function used with a heuristic search algorithm, such as Monte Carlo tree search (MCTS) (Coulom, 2006) or A* search (Hart et al., 1968). However, these approaches are not robust no noise. For example, if irrelevant distractors that were not seen during training are part of a state at test time, the learned world model's prediction may become very unreliable making finding solutions impractical. As a result, these approaches will most likely fail to generalize beyond very carefully controlled settings. Furthermore, when specifying goals with a goal images, noise present in the environment can make goal state identification difficult. For example, if a goal image is specified in a bright room

and the room is currently dark, unless the action space includes controlling the lights, there is no sequence of actions that can reach the goal image.

One approach to solving this problem of noise could be to train the world model on noisy states. However, this suffers from the following drawbacks: (1) The world model is forced to capture noisy transitions, which may require more parameters and hinder performance; (2) Goal state identification still depends on the noise present in the state, which means different goal images must be given based on the noise present in the start state; and (3) It requires that the world model and heuristic function be re-trained if a new type of noise is introduced. On the other hand, another approach would begin with training a world model in a clean environment (i.e. without noise). At test time, if noise is present, the data obtained from training the world model is re-used by adding noise to the states and training an alignment model that maps the noisy states to the clean states. As a result, no new data need be obtained and the world model and heuristic function remain fixed. However, one potential drawback of this approach is that small differences in the denoised state and the clean state could cause compounding errors to occur when applying the world model across many timesteps, also known as model-degradation.

To address these issues, we introduce Stable Planning through Aligned Representations (SPAR). SPAR trains a discrete world model that maps states in a clean environment to a discrete latent state and then trains an alignment model to map noisy states to their discrete latent representation in the clean environment. SPAR exploits the fact that the latent representation is discrete by rounding the output of the alignment model when solving problems with the aim of exactly matching the true discrete latent state. Since discrete world models have been shown to accurately predict next states across thousands of timesteps without model degradation Agostinelli & Soltani (2024), this also enables planning over long horizons without model degradation in the presence of noise. On the other hand, we show that training an alignment model for a world model with a continuous latent representation results in model degradation, even when the continuous model does

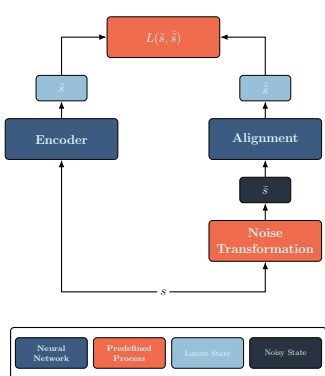

Figure 1: Overview for training the alignment model given a latent space encoder and noise transformation that applies to states to which the world model and heuristic function should be invariant.

not exhibit model degradation in the clean environment. Our approach is summarized in Figure 1.

## 2 RELATED WORK

**Model-based reinforcement learning and planning.** Model-based RL (MBRL) improves sample efficiency by learning a dynamics model and exploiting that model for policy improvement or planning (Sutton, 1990). Dyna architecture interleaves real experience with model-generated rollouts to update value functions and policies (Sutton, 1990). More recent methods couple learned neural models with powerful search procedures. AlphaZero learns value and policy networks and performs Monte Carlo Tree Search (MCTS) in known perfect-information games (Silver et al., 2018), while MuZero achieves superhuman performance without access to the true simulator by learning a latent dynamics model for planning with MCTS (Schrittwieser et al., 2020). PlaNet, Dreamer, and DreamerV2 learn latent world models and optimize policies via latent imagination, but they do not perform explicit test-time search to arbitrary goal states (Hafner et al., 2019b; 2020; 2021). In contrast to policy-only use of models, our setting requires long-horizon search guided by a learned heuristic over representations that support exact state re-identification.

**Learning state representations for search.** A body of work learns symbolic or latent state spaces that enable classical search from pixels. LatPlan learns a discrete binary latent representation with an autoencoder and induces a classical planning model solvable by off-the-shelf planners (Asai & Fukunaga, 2018; Asai et al., 2022). PPGS (Bagatella et al., 2023) learns a continuous latent

space with forward and inverse models, constructs a latent-state graph using threshold-based state re-identification, and then uses uninformed graph search to solve combinatorial puzzles. However, continuous latent rollouts can accumulate error and force frequent re-planning, and uninformed breadth-first search scales poorly (Bagatella et al., 2023). A complementary line learns heuristics to guide search directly from experience. DeepCubeA trains a deep neural network to approximate the cost-to-go via deep approximate value iteration and uses the learned heuristic with A* search to solve the Rubik's Cube and other puzzles, training against a single fixed goal (the canonical solved state) (Agostinelli et al., 2019). DeepCubeAI (Agostinelli & Soltani, 2024) instead learns a discrete world model and a goal-conditioned heuristic defined over binary latent states, rounding each bit at 0.5 eliminates compounding errors in long rollouts and enables state re-identification, supporting efficient search in latent space (Agostinelli et al., 2024b). Our work builds on this discrete-latent planning foundation but addresses robustness to observation variations by learning a lightweight alignment network, leaving the world model and heuristic unchanged.

**Handling visual domain variation.** Domain shift in observations (e.g., lighting, background, camera pose) challenges visual planning systems. Domain randomization broadens the training distribution in simulation so real observations appear as another randomized variant (Tobin et al., 2017a; Sadeghi & Levine, 2017). In robotics, sim-to-real pipelines augment randomization with image-to-image translation, e.g., RCAN maps randomized renders to a canonical domain to improve transfer (James et al., 2019). In RL specifically, data augmentation improves policy robustness (Laskin et al., 2020; Yarats et al., 2021). DARLA learns disentangled representations with beta-VAE pretraining to achieve zero-shot transfer across visual changes before learning to act (Higgins et al., 2017). Beyond training-time robustness, test-time adaptation methods update perception or policies during deployment without labels or rewards, for example PAD for self-supervised policy adaptation (Hansen et al., 2021) and invariance through latent alignment (ILA), which aligns target-domain features to the source distribution without paired data (Yoneda et al., 2022). These approaches, however, generally target policy performance and short-horizon control rather than long-horizon heuristic search that requires exact state re-identification. SPAR complements these lines: we isolate perception shift by learning an alignment network that maps varied observations into a fixed discrete latent space. This preserves the long-horizon stability and exact state re-identification of discrete-latent planning (Agostinelli & Soltani, 2024) while avoiding retraining the dynamics model or heuristic for each new visual change.

## 3 PRELIMINARIES

### 3.1 PATHFINDING

The particular kind of planning problems SPAR addresses are pathfinding problems. A pathfinding problem can be defined as a weighted directed graph (Pohl, 1970), where nodes represent states, edges represent actions that transition between states, weights on the edges represent transition costs, a given state represents the start state, and a given set of states represents the goal. The transition function, $T$, defines how actions transform states, where $s_{t+1} = T(s_t, a_t)$ for action $a_t$, if and only if there exists an edge connecting state $s_t$ to $s_{t+1}$. The transition-cost function, $c(s_t, a_t)$, is the cost of taking action $a_t$ in state $s_t$. While we will use the aforementioned notation throughout the rest of the paper, from a reinforcement learning perspective, a pathfinding problem can also be defined as a deterministic un-discounted Markov decision process (Puterman, 2014). Given a pathfinding problem, the objective is to find a sequence of actions that transforms the start state into a goal state with preference for paths with lowest cost, where the cost of a path is the sum of transition costs. The cost-to-go is the cost of a lowest cost path, also known as a shortest path.

### 3.2 DEEPCUBEAI

**Training** DeepCubeAI (Agostinelli & Soltani, 2024) learns a discrete world model represented as a deep neural network (DNN) (Schmidhuber, 2015) that maps states to binary latent states from an offline dataset of state, action, and next state tuples. The world model is learned using an encoder to map states to discrete latent states, a decoder that maps discrete latent states to states, and a world model that maps discrete latent states and actions to next states. The encoder and decoder are trained

to minimize the reconstruction error between the input to the encoder and the output of the decoder. Simultaneously, the encoder and world model are trained to accurately predict the next state.

After learning, the outputs of the discrete world model are rounded, which prevents model degradation when errors are less than 0.5. Using the learned world model, DeepCubeAI then learns a heuristic function represented as a deep Q-network (DQN) (Mnih et al., 2015), using Q-learning (Watkins & Dayan, 1992) to estimate the cost-to-go when starting from a given state and taking a given action. Training data is obtained by using the world model to generate new experiences from the offline dataset. As a result, the heuristic function takes latent states as input.

**Solving Pathfinding Problems**    After training, a particular pathfinding problem is given to DeepCubeAI in the form of a start state image and a goal image. DeepCubeAI then encodes the start state and goal state into the discrete latent space and uses the learned world model to perform heuristic search in the latent space, guided by the learned heuristic function, to find a path from the latent start state to the latent goal state. The heuristic search algorithm checks whether a latent state encountered during search is a goal state by checking if all bits match the latent goal state. Since the world model is represented as a DNN, using it for heuristic search can be computationally expensive. Therefore, Q* search (Agostinelli et al., 2024b), a variant of A* search (Hart et al., 1968) that uses a DQN as the heuristic function, is used to find paths. This is because, with respect to the size of the action space, the number of applications of the transition function (the world model, in this case) remains constant for each iteration in the case of Q* search as opposed to growing linearly in the case of A* search.

## 4    METHODS

SPAR assumes a trained encoder that maps states to latent states and a trained world model that maps latent states and actions to next latent states. Given a pretrained world model, our goal is to train an alignment model that maps noisy states to their corresponding discrete latent states in the clean environment. We train this model with supervised learning on a dataset of $(\bar{s}, \tilde{s})$ pairs (noisy states paired with discrete latent representation of the corresponding clean states) minimizing mean squared error (MSE) between the model output and the target latent state. To gather training data, we reuse the dataset originally used for the world model, applying some transformations with random intensities, that introduces perturbations to which the world model and heuristic function should be invariant. This transformation can be applied either to states in the offline dataset or to new states sampled from the environment, if the original dataset is not available.

### 4.1    DISCRETE WORLD MODEL

We map pixel observations into a discrete latent space and learn transitions within it. Let $E$ be the encoder, $D$ the decoder, and $T$ the transition model. Encoder outputs pass through a logistic layer and are rounded to $\{0,1\}$, and a straight-through estimator (Bengio et al., 2013) is used to enable gradient-based training.

Given an offline dataset of transitions $\{(s_t^{(i)}, a^{(i)}, s_{t+1}^{(i)})\}_{i=1}^N$, we optimize two losses. First, a reconstruction loss that encourages meaningful encodings and decodings, as given in Equation 1.

$$L_r(\theta) = \frac{1}{N} \sum_{i=1}^{N} \left[ \frac{1}{2} \left\| s_t^{(i)} - D\big(E(s_t^{(i)})\big) \right\|_2^2 + \frac{1}{2} \left\| s_{t+1}^{(i)} - D\big(E(s_{t+1}^{(i)})\big) \right\|_2^2 \right]. \tag{1}$$

Second, we define a transition loss that couples the encoder and transition model. Let $\tilde{s}_t^{(i)} = E(s_t^{(i)})$ and $\tilde{s}_{t+1}^{(i)} = E(s_{t+1}^{(i)})$ be the continuous encoder outputs, and let $r(\cdot)$ denote elementwise rounding to $\{0, 1\}$. The transition model predicts the next latent state $\hat{\tilde{s}}_{t+1}^{(i)} = T\big(\tilde{s}_t^{(i)}, a^{(i)}\big)$. We use a symmetric loss with a stop-gradient operator $\text{sg}(\cdot)$ applied to one side of each term to stabilize encoder-transition training, as given in Equation 2.

$$L_m(\theta) = \frac{1}{N} \sum_{i=1}^{N} \left[ \frac{1}{2} \left\| r(\tilde{s}_{t+1}^{(i)}) - \text{sg}\big(r(\hat{\tilde{s}}_{t+1}^{(i)})\big) \right\|_2^2 + \frac{1}{2} \left\| \text{sg}\big(r(\tilde{s}_{t+1}^{(i)})\big) - r(\hat{\tilde{s}}_{t+1}^{(i)}) \right\|_2^2 \right]. \tag{2}$$

## 4.2 ALIGNMENT MODEL

Given pretrained models $(E, T, D)$ learned in a clean setting, we train an alignment network $A$ that maps a visually transformed observation $\bar{s}$ to the discrete latent of the corresponding clean observation $s$. Training data are constructed as pairs of (variant observation, base observation) by aligning variant frames with their corresponding base frames at the same time step within each episode.

Supervised learning can be used to train $A$ by minimizing the MSE between its real-valued outputs and the discrete latent targets produced by the encoder. Concretely, we define $\hat{\tilde{s}}^{(i)} = A(\bar{s}^{(i)})$ and $\tilde{s}^{(i)} = r(E(s^{(i)}))$, and minimize the loss in Equation 3, where $\phi$ denotes the parameters of $A$. At inference time, we obtain discrete alignment codes by rounding the outputs $A(\bar{s})$ elementwise to $\{0, 1\}$.

$$L_A(\phi) = \frac{1}{N} \sum_{i=1}^{N} \left\| \hat{\tilde{s}}^{(i)} - \tilde{s}^{(i)} \right\|_2^2 \tag{3}$$

**Visual variations.** To probe visual robustness, we apply a series of perturbations to the images of Rubik's Cube and Sokoban. These include pre-render (background), object-render (lighting, color, geometry), and post-render stages. Additionally, we apply object-level and background transformations to Rubik's Cube, such as color shifts and calibrations, sticker wear, material finish, camera viewpoint changes and zoom, and a physically-motivated directional lighting model (Tobin et al., 2017b; Zhong et al., 2020; Shorten & Khoshgoftaar, 2019; Hendrycks & Dietterich, 2019; Da et al., 2025). Backgrounds used for Rubik's Cube included images from CIFAR dataset (Krizhevsky et al., 2009), selected randomly. These transformations were applied with parameters sampled per-frame to cover a wide range of observation changes. The transformations are explained in Appendix D along with their example figures. However, we use combinations of these individual transformations for Rubik's Cube and, both combinations and individual augmentations for Sokoban. Figure 2 is an example of the data in our dataset.

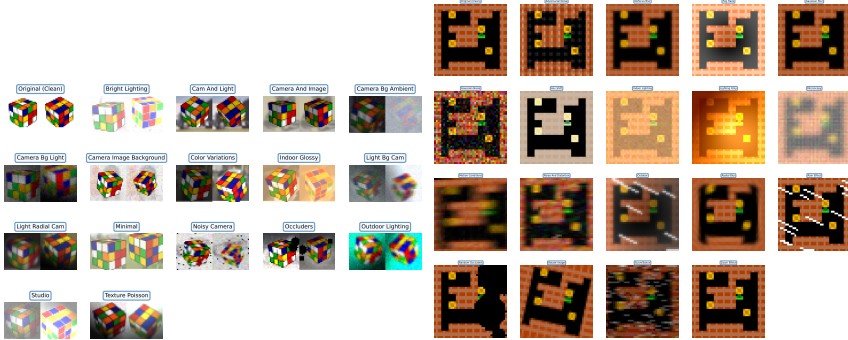

Figure 2: Visualization of sample data from the datasets used for training the alignment model for Rubik's Cube (on the left) and Sokoban (on the right).

## 5 EXPERIMENTS

We evaluate SPAR, isolating the role of discrete latents and the alignment network in long-horizon planning. Below we detail the experimental setup, models, training procedures, and evaluation framework. For, world model training and heuristic learning, we use the a similar setup as DeepCubeAI with the same underlying logic and implementation as their publicly available codebase (Soltani, 2025).

### 5.1 EXPERIMENTAL SETUP

**Domain.** We consider visual path-finding domains with deterministic dynamics and discrete action spaces. Our primary domain is $3 \times 3 \times 3$ Rubik's Cube. We generate offline datasets of episodes by

random action policies constrained to valid moves. We use quarter-turn metric (QTM) to measure solution cost, where each 90-degree face turn has unit cost. The action space has 12 actions, one for each face and direction. States are represented as $32 \times 32$ RGB images.

**Offline datasets.** For training the world model and heuristic function, we generate an offline dataset of 10,000 episodes using action sequences of 30 steps, sampled uniformly at random from the valid move set. Then we use 90% of the dataset for training and 10% for validation. Start states are sampled by applying 100–200 inverse moves from the canonical solved configuration, ensuring diverse but reachable states. Moreover, we generate a dataset of 100 episodes and 10,000 steps for comparing rollout stability between discrete and continuous models. For training the alignment model, we generate a dataset of 10,000 episodes and 30 steps for per each visual variant of the environment by applying random transformations to states in the original dataset, and we pair each transformed state with its corresponding base state. However, it is possible to use a new dataset that includes visual variations and clean images that are not present in the original dataset.

## 5.2 Model Architectures

Encoder, decoder, transition model, and heuristic model follows DeepCubeAI's architecture. Details of these models are given in Appendix A. For the alignment model, both for Rubik's Cube and Sokoban we use a convolutional residual architecture. The input is a 6-channel tensor which first passes through a batch normalization layer, followed by two convolutional layers. The first layer maps $6 \rightarrow 64$ channels with a $5 \times 5$ kernel, stride 2, and padding 2. The second layer maps $64 \rightarrow 36$ channels with a $3 \times 3$ kernel, stride 1, and padding 1. Both layers use batch normalization, and ReLU non-linearities. The convolutional head is followed by a residual block consisting of 12 residual blocks of 38 channels, each with batch normalization and ReLU activation. The output is then flattened and passed through a two-layer MLP of size $9216 \rightarrow 10000 \rightarrow 400$. Batch normalization and ReLU are applied after, and a sigmoid activation is used at the final layer to produce a 400-dimensional output aligned to the encoder's latent space.

## 5.3 Training Procedures

**Discrete world model.** We train an encoder $E$, transition model $T$, and decoder $D$ on base images. For Rubik's Cube, $E$ and $D$ were fully connected networks. $E$ outputs a 400-dimensional vector, is then passed through a logistic function and rounded to $\{0, 1\}^{400}$. We train $E$ and $T$ jointly, minimizing the reconstruction and transition objectives in Equations 1 and 2 with a scheduler that increases the transition model's loss weight from $1 \times 10^{-3}$ over time until both the encoder's loss and transition model's loss have an equal weight of 0.5. We use Adam optimizer (Kingma & Ba, 2014) with learning rate $10^{-3}$, weight decay $10^{-5}$, and batch size 100. We use the same architecture for the continuous model. However, we do not round the outputs of the encoder and the transition model in this case.

**Alignment model.** Given some pre-trained models $(E, T, D)$ trained on clean observation $s$, we train an alignment network $A$ to map noisy observation $\bar{s}$ to clean environment's discrete latent $r\big(E(s)\big)$. We use the same optimizer and scheduler as the discrete world model training, and loss defined in Equation 3.

**Heuristic learning.** We train a goal-conditioned DQN over discrete latent using Q-learning with targets defined by the world model, and assuming unit transition costs for each action taken. Training data is synthesized by random walks in latent space via $T$. Actions for data generation are sampled with a Boltzmann distribution over current Q-values. Similar to to priror work, we maintain a target network and update it if greedy best-first rollouts evaluation of the DQN being trained, is improved.

## 5.4 Evaluation Framework

**Planning performance.** We assess planning by mapping start and goal observations to discrete latents, using the alignment model. Then we run Q* search in discrete latent space with exact bitwise goal tests. We report the percentage of problems solved and the cost of found solutions in Table 1, and the results for each visual transformation in Table 2. We also compare the results

to a greedy policy, which uses the learned heuristic to select the action with the lowest estimated cost-to-go at each step for 100 iteration.

Table 1: Comparison of SPAR (ours) with a greedy policy and DeepCubeAI on different variants of the observations.

| DOMAIN | OBSERVATION | SOLVER | LEN | NODES | SECS | NODES/SEC | SOLVED |
|---|---|---|---|---|---|---|---|
| Rubik's Cube | Clean | SPAR | 23.61 | 1.91E+05 | 6.18 | 3.10E+04 | 100% |
| | | DeepCubeAI | 23.61 | 1.91E+05 | 5.96 | 3.20E+04 | 100% |
| | | Greedy Policy | – | – | – | – | 0% |
| | Augmented | SPAR | 23.64 | 1.91E+05 | 6.22 | 3.08E+05 | 89.39% |
| | | DeepCubeAI | – | – | – | – | 0% |
| | | Greedy Policy | – | – | – | – | 0% |
| | Real-World | SPAR | 23.61 | 1.90E+05 | 6.21 | 3.10E.51 | 50% |
| | | DeepCubeAI | – | – | – | – | 0% |
| | | Greedy Policy | – | – | – | – | 0% |
| Sokoban | Clean | SPAR | 33.12 | 3.30E+03 | 2.64 | 1.25E+03 | 100% |
| | | DeepCubeAI | 33.12 | 3.30E+03 | 2.62 | 1.38E+03 | 100% |
| | | Greedy Policy | – | – | – | – | 41.9% |
| | Augmented | SPAR | 34.55 | 3.31E+03 | 2.65 | 1.25E+03 | 96.45% |
| | | DeepCubeAI | – | – | – | – | 0% |
| | | Greedy Policy | – | – | – | – | 0% |

**Model Performance.** To determine the performance of the alignment model, we evaluate both models on 100 sequences of 10,000 steps each, with actions selected uniformly at random. For every step, we record the ground-truth image, advance one step in the latent space, and reconstruct the observation using the decoder. We also compare against a continuous variant that shares the same architecture and training procedure as the discrete model but omits discretization. Results, shown in Figure 4, indicate that the continuous model accumulates substantial error for Rubik's Cube. Figure 3 shows the mean squared error across 17 transformations of the environment from Rubik's Cube domain where the continuous model exhibits degradation, whereas the discrete model remains accurat. The exhaustive reconstruction MSE plots for Rubik's Cube and Sokoban are available in Appendix C.

Figure 3: MSE for discrete vs. continuous alignment models across Rubik's Cube variants over 10,000 steps. The label "predicting next state" refers to the training where the given current state and action, world model predicts next state, and the objective of alignment model is, given a visually transformed current state, to minimize the reconstruction loss between prediction of next state and the next state observed in a clean environment.

**Generalization to Unseen Variations and Real Images.** To assess generalization, capabilities of SPAR, we evaluate on compositions and random intensities of transformations that were not present during alignment training. Moreover, we use the same trained alignment model to eval-

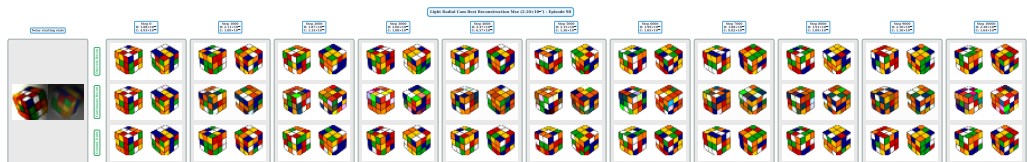

Figure 4: Reconstruction of the alignment model's outputs starting from a noisy observation.

uate our framework on real-world data. We use photos of a Rubik's Cube taken in real-world in different lighting conditions and backgrounds. We take two photos of Rubik's Cube in a similar orientation to the offline data. These two photos are then resized to $32 \times 32$ and concatenated. The prediction of the alignment model is then passed to the pretrained decoder to get the reconstruction of the discrete encoding (Figure 6). We compare the The discrete alignment retains strong performance when perturbations respect structural cues (e.g., moderate color shifts and lighting changes) and degrades gracefully when cues are heavily occluded or geometry is distorted. Figure 5 and Appendix F provide example visualizations of variants that the search component could not find a solution for them. In cases where the alignment model's output match the outputs of the encoder applied to the corresponding state in the simulation, the planning component finds a path to given goal.

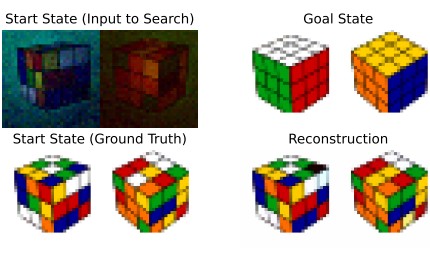

Figure 5: An example of an augmentation that SPAR fails to accurately predict the discrete latent code, and therefore is not solved during planning.

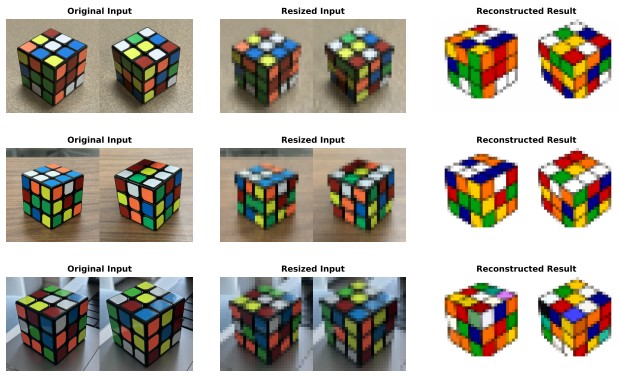

Figure 6: Applying SPAR to a real-world Rubik's Cube image with differing lighting/background.

## 6 DISCUSSION

Discrete latents act as an error-correcting bottleneck for planning. Rounding removes small deviations, enabling deep rollouts without drift and precise state matching. Decoupling the encoder from dynamics learning yields practical benefits. When new variations are introduced to the environment, only re-training the alignment model is enough to leverage the benefits of the discrete world model and its quality perseverance and capabilities when used with heuristic learning and planning, while the world model and heuristic remain intact. Empirically, this substantially reduces retraining time, from days need to train a heuristic function down to a few minutes or hours, and preserves planning competence under varying visual conditions.

## 7 LIMITATIONS AND FUTURE WORK

SPAR separates perception alignment from planning, offering a promising foundation for robust decision making. Future work can focus on extending its adaptability to richer forms of variation,

enabling online learning, linking with foundation and symbolic models, and validating its utility in real-world domains. We first summarize key limitations and failure modes, and then outline directions to address them.

**Reliance on paired data.** A potential limitation is the reliance on paired data (i.e., a "noisy" observation and its corresponding "clean" observation) for training the alignment network. While feasible in simulation, acquiring such perfectly paired data in the real world can be challenging. To relax this assumption, future work can explore: (i) weakly supervised or unpaired objectives. (ii) domain adaptation and test-time adaptation to update the alignment model without labels.

**Failure cases.** SPAR can fail when perturbations erase task-relevant structure (e.g., severe occlusion) or when observations fall far off the training distribution, leading to incorrect latent rounding. Detecting and mitigating these cases is an open direction: confidence estimates over discrete latents, agreement checks across augmentations, abstention or fallback policies, adversarially generated augmentations, and robust objectives could reduce brittleness and improve reliability.

**Beyond deterministic, fully observable settings.** Our experiments consider deterministic dynamics with fully observable states. Extending SPAR to stochastic and partially observable environments invites several avenues: (i) maintain a belief over discrete latents (e.g., via recurrent state) and train the alignment network to output observation likelihoods rather than a single code. (ii) Plan in belief space using POMDP or over a compact belief parameterization. (iii) Use temporal windows and predictive state representations to disambiguate aliased observations. These extensions would broaden applicability to real-world robotics and control.

**Integration with Large-Scale Models.** In the same vein, we envision using large vision-language models (VLMs) or large language models (LLMs) to guide planning. Complementarily, SPAR's discrete latents can be mapped to a predicate inventory and interfaced with formal goal languages specifying what must (or must not) hold, and a solver validates these targets before SPAR plans (Agostinelli et al., 2024a). For interpretability and reuse, inductive logic programming (ILP) can be used along with SPAR's canonical states to provide human-readable rationales (Agostinelli et al., 2022). Incorporating multi-modal planning capabilities, e.g., specifying goals in natural language, would broaden SPAR's applicability.

**Broader Deployment Domains.** Finally, applying SPAR to real-world sequential decision-making problems is an important direction. In robotics, an agent often encounters varying lighting, backgrounds, etc. SPAR could provide a fast way to recalibrate the robot's perception without retraining its policy or dynamics model each time the environment changes (Longhini et al., 2024; Nutalapati et al., 2022). Fast adaptation of perception without full retraining aligns with test-time adaptation results in robotics and perception (Park et al., 2024; Piriyajitakonkij et al., 2024). Coupling SPAR with classic robot planning algorithms or continuous control techniques may be necessary to handle continuous action spaces, but the concept of a discrete latent adapter remains useful.

In summary, SPAR takes a significant step toward robust model-based planning by cleanly separating the concerns of dynamics and perception. By anchoring different visual observations to a common discrete representation, it achieves long-horizon stability with minimal retraining.

## 8 CONCLUSION

Our study across different transformation families shows that this separation is practical: an alignment model alone suffices to recover planning performance while cutting retraining time by at least 95% while having the same pathfinding capabilities as DeepCubeAI. Conceptually, SPAR complements two major threads: (i) model-based RL with latent planning, where discrete latents and imagination have driven strong performance, and (ii) robustness and sim-to-real pipelines that map diverse observations into a canonical domain. By aligning into a discrete latent space and then planning with exact goal tests, SPAR maintains long-horizon stability while adapting rapidly to observation changes.

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

## A  APPENDIX: MODEL ARCHITECTURE

Here, we detail the neural network architectures used for encoder, decoder, transition model, and heuristic model.

For Rubik's Cube, We use fully connected networks for encoder and decoder with one hidden layer, and output a 400-dimensional vector. Similarly, decoder uses fully connected networks with a linear (identity) function as the final activation. For the transition model, we use four fully connected layers of size $500 \rightarrow 500 \rightarrow 500 \rightarrow 400$ with batch normalization and ReLU, except for the last layer where we do not use batch normalization with logistic outputs. Actions are one-hot encoded and concatenated to the input state and passed to the neural network. For heuristic function, we use a deep Q-network (DQN) (Mnih et al., 2013) with a fully connected residual network. The input is the concatenation of the discrete latent representations of current state and goal state. A two-layer MLP maps $800 \rightarrow 5000 \rightarrow 1000$ with batch normalization and ReLU after each layer. This is followed by

4 residual blocks of width 1000, each block having batch normalization and ReLU activation. At the final layer, a linear activation outputs Q-values for all actions ($1000 \rightarrow 12$).

For Sokoban, we use a two-layer convolutional encoder with $2 \times 2$ kernels, stride 2, and zero paddings, mapping channels `chan_in` $\rightarrow 16 \rightarrow 16$. Batch normalization and ReLU are applied after the first layer. The second layer uses a sigmoid activation. The resulting feature map has spatial size $10 \times 10$ with 16 channels and is flattened to a 1600-dimensional vector ($10 \times 10 \times 16$). For transition model, actions are one-hot encoded as channel planes and concatenated with the current latent feature maps along the channel dimension. The transition model is a 3-layer $3 \times 3$ convolutional network with stride 1 and padding 1 (`number of actions` $+ 16) \rightarrow 32 \rightarrow 32 \rightarrow 16$. The first two layers use batch normalization and ReLU, and the final layer omits batch normalization and uses a sigmoid output. The decoder mirrors the encoder with two transposed $2 \times 2$ convolutions of stride 2, mapping $16 \rightarrow 16 \rightarrow 16$ channels (batch norm + ReLU on the first, sigmoid on the second) to upsample back to the original spatial resolution. A final $1 \times 1$ convolution maps $16 \rightarrow$ `chan_in` with a linear (identity) activation to reconstruct the observation. Similar to Rubik's Cube, the DQN input is the concatenation of the encoded current state and goal state, giving a $2 \times 1600 = 3200$-dimensional vector. A two-layer MLP maps $3200 \rightarrow 5000 \rightarrow 1000$ with batch normalization and ReLU after each layer. This is followed by 4 fully connected residual blocks of width 1000, each using batch normalization and ReLU. A final linear head outputs Q-values for all actions ($1000 \rightarrow$ `number of actions`).

## B APPENDIX: ADDITIONAL ROLLOUTS AND VISUALIZATIONS

In this appendix we provide supplementary rollout visualizations for the discrete and continuous alignment models for the Rubik's Cube and Sokoban domains. Each figure block below shows: (a) the reconstructions of discrete latent representation (on the first row), (b) the reconstructions of continuous latent representation (on the second row), and (c) the ground truth clean observations (on the third row). All images show reconstructed frames from the alignment model's latent predictions with inputs given in the left panels. The output of the alignment model is then given to decoder for visualizing the reconstruction.

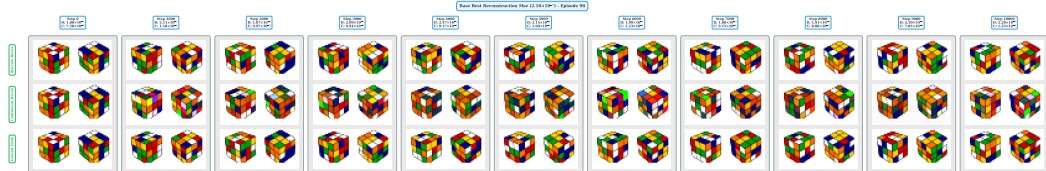

Figure 7: Reconstructions of rolling out the discrete and continuous alignment models for the base (clean) setup for Rubik's Cube.

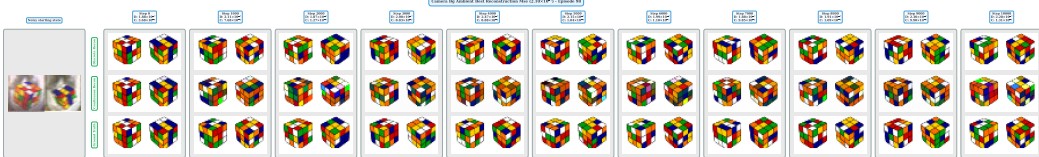

Figure 8: Reconstructions of rolling out the discrete and continuous alignment models for the Ambient Background + Camera perturbation for Rubik's Cube.

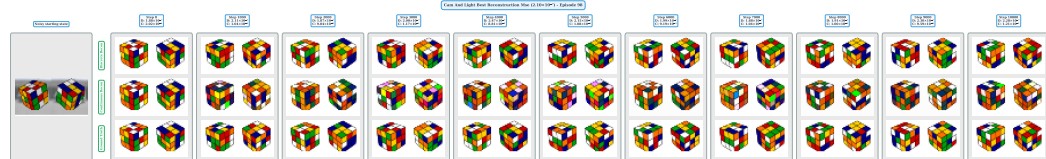

Figure 9: Reconstructions of rolling out the discrete and continuous alignment models for the Camera + Lighting perturbation for Rubik's Cube.

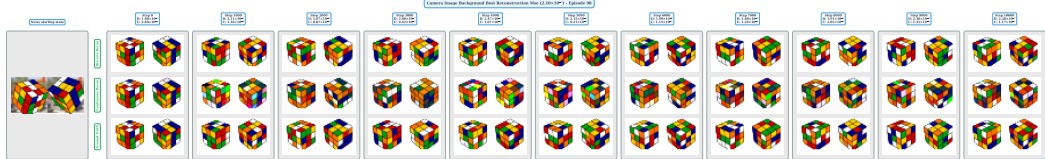

Figure 10: Reconstructions of rolling out the discrete and continuous alignment models for the Image Background perturbation for Rubik's Cube.

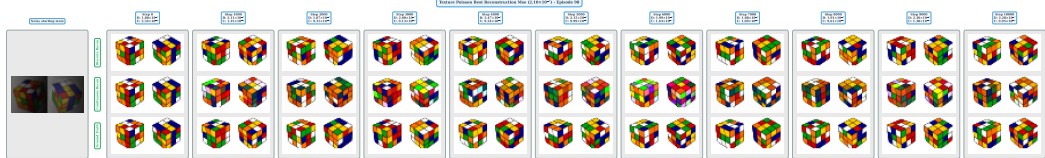

Figure 11: Reconstructions of rolling out the discrete and continuous alignment models for the Textured Background + Poisson Noise perturbation for Rubik's Cube.

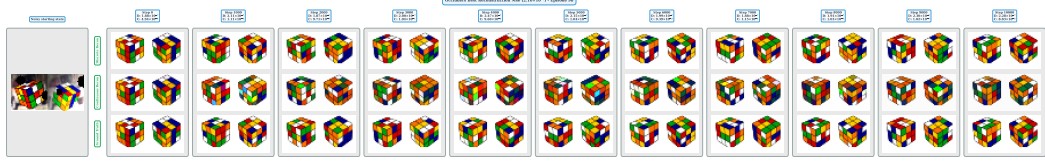

Figure 12: Reconstructions of rolling out the discrete and continuous alignment models for the Occlusions perturbation for Rubik's Cube.

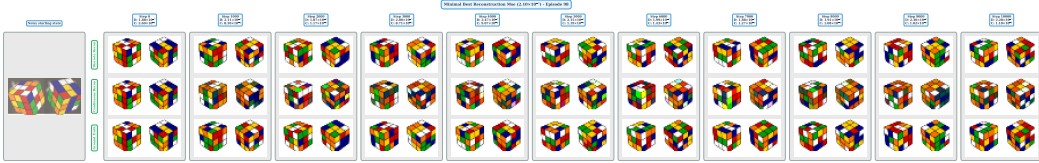

Figure 13: Reconstructions of rolling out the discrete and continuous alignment models for the Minimal Augmentations perturbation for Rubik's Cube.

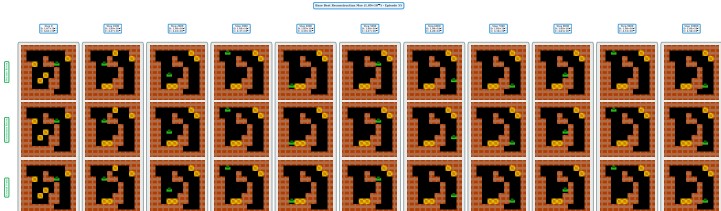

Figure 14: Reconstructions of rolling out the discrete and continuous alignment models for the base (clean) setup for Sokoban.

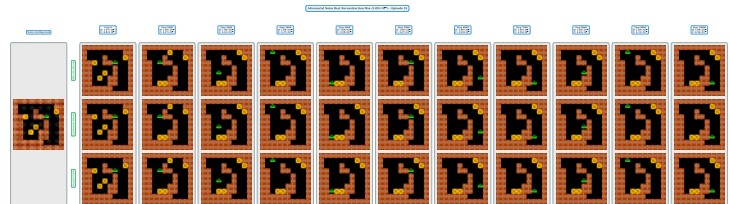

Figure 15: Reconstructions of rolling out the discrete and continuous alignment models for the Adversarial Noise perturbation for Sokoban.

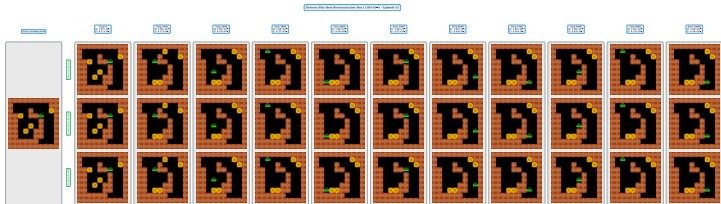

Figure 16: Reconstructions of rolling out the discrete and continuous alignment models for the Defocus Blur perturbation for Sokoban.

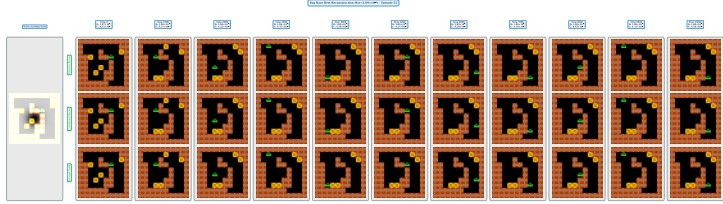

Figure 17: Reconstructions of rolling out the discrete and continuous alignment models for the Fog Haze perturbation for Sokoban.

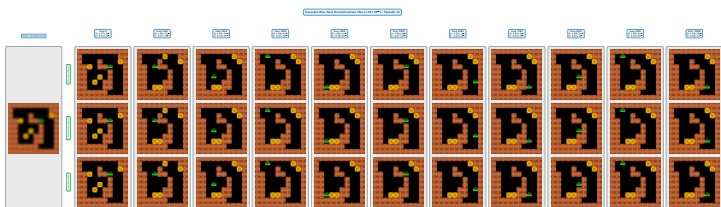

Figure 18: Reconstructions of rolling out the discrete and continuous alignment models for the Gaussian Blur perturbation for Sokoban.

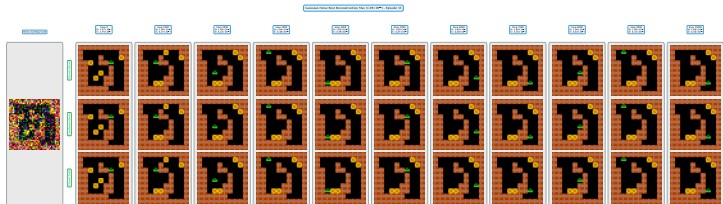

Figure 19: Reconstructions of rolling out the discrete and continuous alignment models for the Gaussian Noise perturbation for Sokoban.

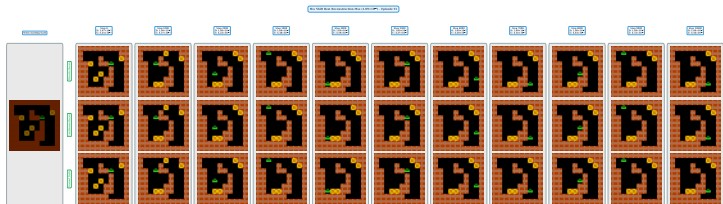

Figure 20: Reconstructions of rolling out the discrete and continuous alignment models for the HSV Shift perturbation for Sokoban.

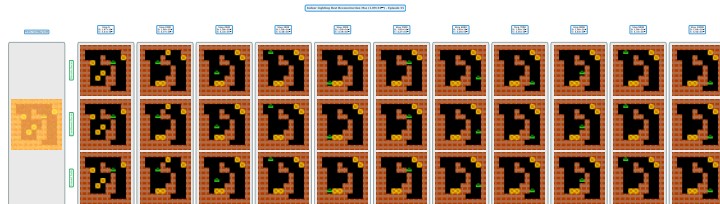

Figure 21: Reconstructions of rolling out the discrete and continuous alignment models for the Indoor Lighting perturbation for Sokoban.

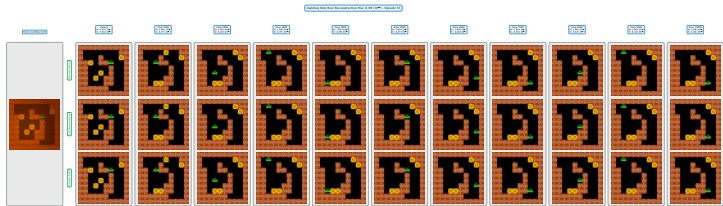

Figure 22: Reconstructions of rolling out the discrete and continuous alignment models for the Lighting Only perturbation for Sokoban.

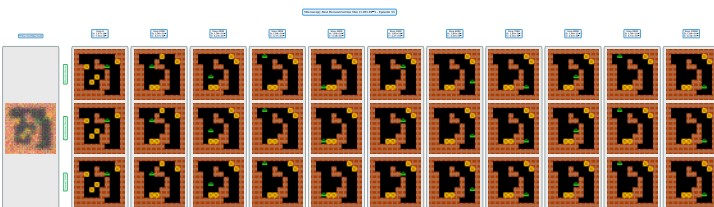

Figure 23: Reconstructions of rolling out the discrete and continuous alignment models for the Microscopy perturbation for Sokoban.

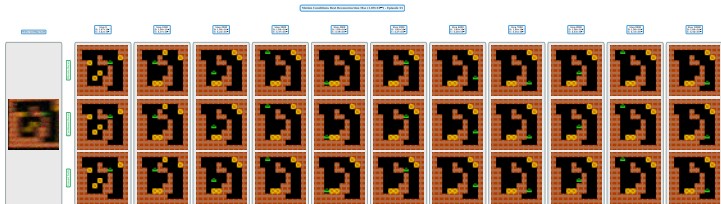

Figure 24: Reconstructions of rolling out the discrete and continuous alignment models for the Motion Conditions perturbation for Sokoban.

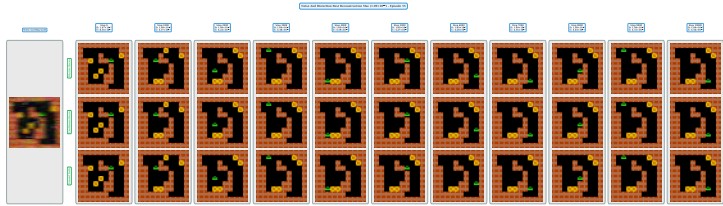

Figure 25: Reconstructions of rolling out the discrete and continuous alignment models for the Noise and Distortion perturbation for Sokoban.

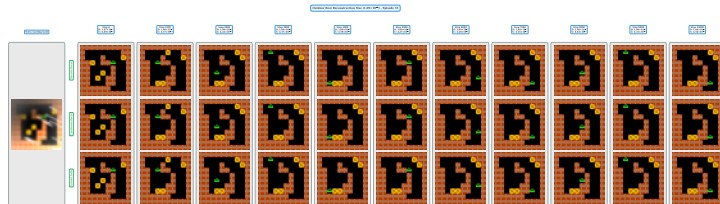

Figure 26: Reconstructions of rolling out the discrete and continuous alignment models for the Outdoor perturbation for Sokoban.

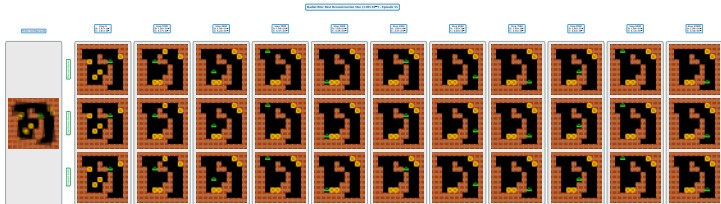

Figure 27: Reconstructions of rolling out the discrete and continuous alignment models for the Radial Blur perturbation for Sokoban.

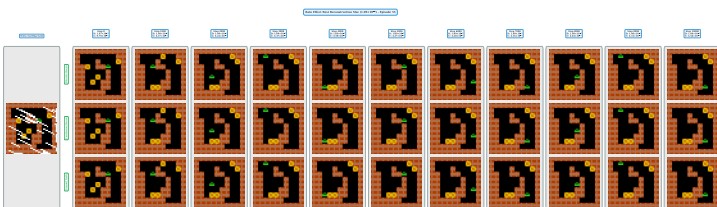

Figure 28: Reconstructions of rolling out the discrete and continuous alignment models for the Rain Effect perturbation for Sokoban.

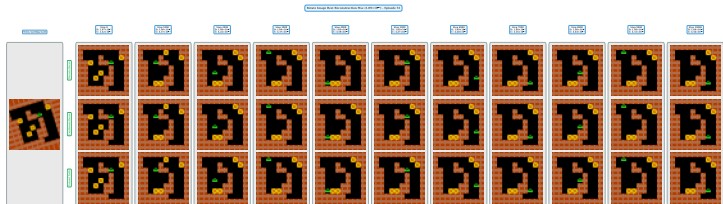

Figure 29: Reconstructions of rolling out the discrete and continuous alignment models for the Rotate Image perturbation for Sokoban.

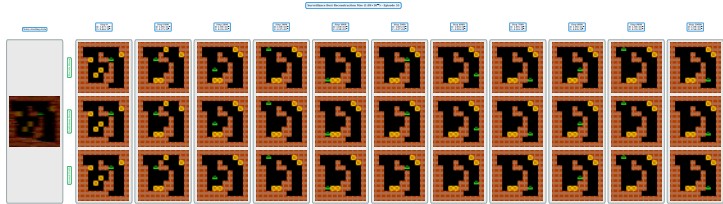

Figure 30: Reconstructions of rolling out the discrete and continuous alignment models for the Surveillance perturbation for Sokoban.

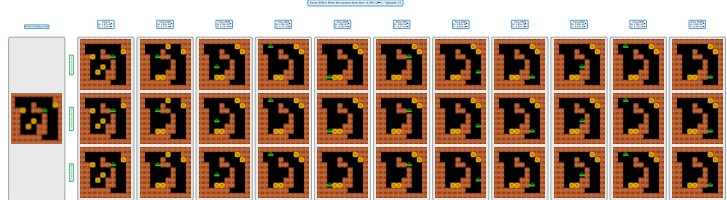

Figure 31: Reconstructions of rolling out the discrete and continuous alignment models for the Zoom Effect perturbation for Sokoban.

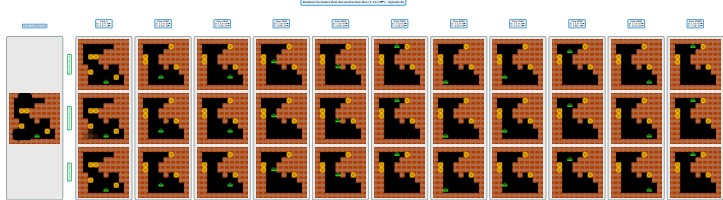

Figure 32: Reconstructions of rolling out the discrete and continuous alignment models for the Random Occluders perturbation for Sokoban.

## C  APPENDIX: MODEL COMPARISON PLOTS

We compile every reconstruction MSE model-comparison plot generated for the Rubik's Cube and Sokoban benchmarks. Each subfigure contrasts the discrete world model, its continuous counterpart, and the aligned variants under a specific appearance change or aggregation. Figure 33 shows the Rubik's Cube plots and Figure 34 shows the Sokoban plots.

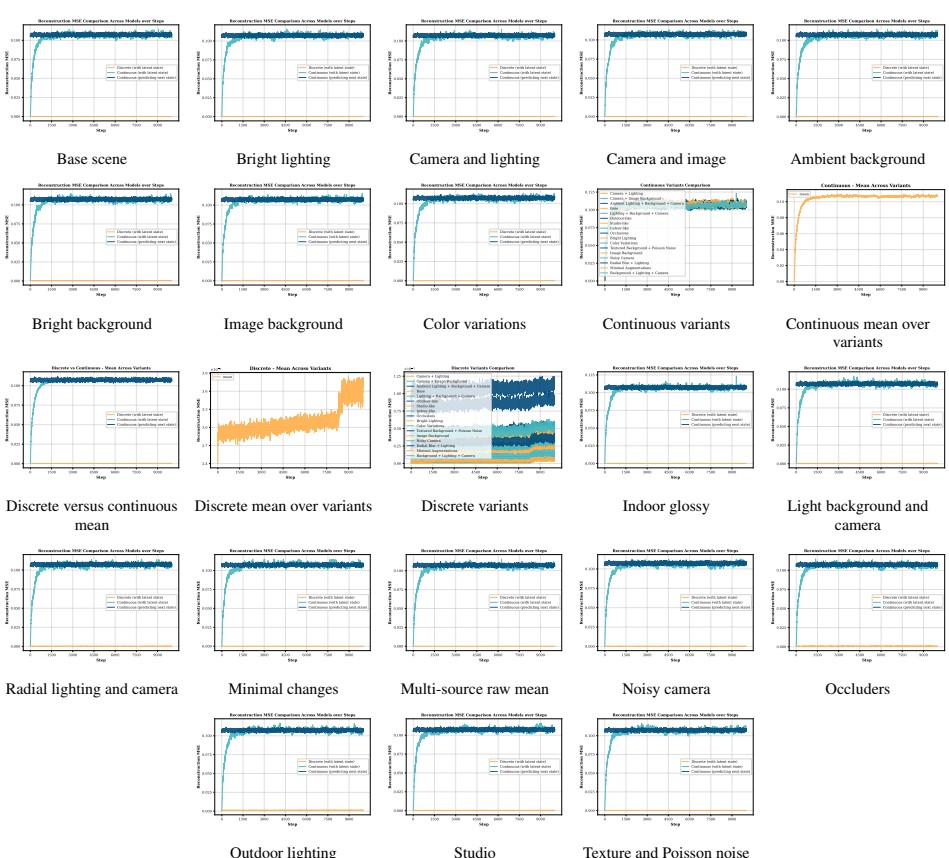

Figure 33: Rubik's Cube reconstruction MSE comparisons across lighting, camera, background, texture, and occlusion variations, showing discrete, continuous, and aligned model curves where lower values reflect better reconstruction fidelity.

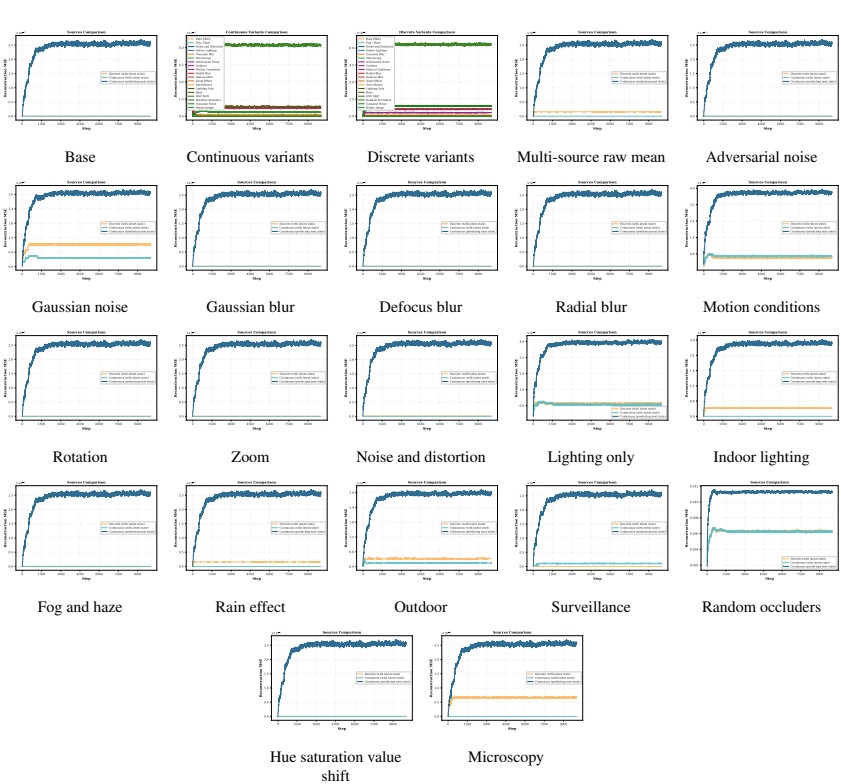

Figure 34: Sokoban reconstruction MSE comparisons covering lighting, blur, motion, noise, viewpoint, weather, and occlusion changes, presenting discrete, continuous, and aligned model curves where lower values indicate stronger reconstruction.

# D APPENDIX: ENVIRONMENTAL PERTURBATION AUGMENTATIONS

This appendix formalizes the augmentations used to simulate environmental perturbations in the Rubik's cube environment. For each augmentation, we state the governing principle, outline the methodology of application, specify controlling parameters and their qualitative effects. We denote an image by $I : \Omega \subset \mathbb{R}^2 \to [0,1]^3$ with per-channel intensities in $[0,1]$. Elementwise operations act per pixel and per channel unless noted. $\mathrm{clip}(\cdot)$ truncates to $[0,1]$. Convolution is $*$. Homogeneous image coordinates are $\tilde{\boldsymbol{x}} = (x, y, 1)^\top$.

**Directional Surface Lighting (Lambertian)** Per-surface intensity follows a Lambertian model augmented with ambient and back-face floor terms. Let $\boldsymbol{n}$ be the surface normal and $\boldsymbol{\ell}$ a unit light direction. The shading field is $S = \mathrm{clip}\left(A + (1 - A)\max(0, \boldsymbol{n}\cdot\boldsymbol{\ell}) + s\right)^\gamma$, and the observed color is $\mathrm{clip}(S \odot \boldsymbol{c})$, where $A$ is ambient, $s$ a subsurface/back-face term, $\gamma > 0$ a contrast control, and $\boldsymbol{c}$ the base reflectance. Azimuth and elevation set $\boldsymbol{\ell}$. This augmentation represents directional illumination, self-shadowing, and highlight contrast on faceted objects.

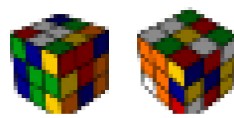

Figure 35: $A = 0.2, \kappa = 0.6, \gamma = 1.5, s = 0.1, azdeg = 315, altdeg = 45$

**Random Background Chromaticity** To diversify global backgrounds across frames, the constant color is drawn from a distribution, typically $\boldsymbol{c} \sim \mathcal{U}([0,1]^3)$. The rendering pipeline applies the sampled $\boldsymbol{c}$ before foreground compositing, inducing broad variation in global tone and contrast. Parameters are the sampling law and any constraints on hue or saturation. This models uncontrolled ambient lighting and incidental background coloration across capture sessions.

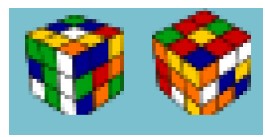

Figure 36: $\boldsymbol{c} \sim \mathcal{U}([0,1]^3)$

**Procedural Background Textures** A parametric texture $B(\boldsymbol{x})$ (e.g., grids, dots, stripes, noise, crosshatch, sinusoids) is synthesized in image coordinates and alpha-composited behind the object, $I' = (1 - \alpha)I + \alpha B$. The construction of $B$ governs spatial frequency, orientation, regularity, and contrast. Parameters include the texture family, spatial density, tint, and opacity $\alpha \in [0,1]$. This augmentation reproduces structured backgrounds such as patterned walls or tabletops that introduce periodic distractors and clutter.

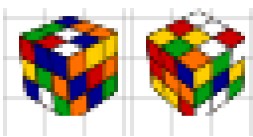

Figure 37: $B(\boldsymbol{x}) = grid, color = gray, density = 1, \alpha = 0.3$

**Hue-Saturation-Value Adjustment** Chromatic attributes are modified by mapping RGB to HSV, applying a hue rotation and multiplicative scaling of saturation and value, then mapping back. Writing $(h, s, v) \mapsto (h', s_{t+1}, v')$ with $h' = (h + \Delta_h) \bmod 1$, $s_{t+1} = \mathrm{clip}(\lambda_s s)$, $v' = \mathrm{clip}(\lambda_v v)$ alters

perceived colorfulness and brightness while preserving relative luminance structure. The parameters are $\Delta_h$, $\lambda_s \geq 0$, and $\lambda_v \geq 0$. This simulates camera white-balance shifts, colored illumination, and post-processing tints.

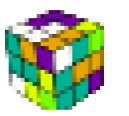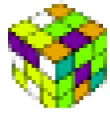

Figure 38: $\Delta_h = 0.12, \lambda_s = 1.25, \lambda_v = 1.12$

**Contrast and Brightness**  A global affine photometric transform, $I' = \mathrm{clip}(\alpha I + \beta)$, adjusts contrast via the gain $\alpha > 0$ and brightness via the offset $\beta \in \mathbb{R}$. The method is applied uniformly per pixel and channel, followed by clipping. This models exposure settings and tone-curve changes typical of automatic image pipelines.

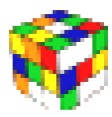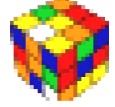

Figure 39: $\alpha = 1.25, \beta = 0.1$

**Exposure via Gamma**  Nonlinear tone mapping with gamma correction redistributes mid-tones while preserving black/white after clipping: $I' = \mathrm{clip}(I^{1/\gamma})$. The parameter $\gamma > 0$ brightens ($\gamma < 1$) or darkens ($\gamma > 1$) mid-range intensities. The augmentation captures low-light and short exposure regimes.

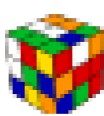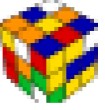

Figure 40: $\gamma = 1.5$

**Color Temperature Shift**  A chromatic bias along the red-blue axis emulates warm/cool illumination. In RGB, a simple model biases red and blue in opposite directions, e.g., $I'_R = \mathrm{clip}(I_R + \delta)$, $I'_B = \mathrm{clip}(I_B - \delta)$, leaving green largely unchanged. The scalar $\delta$ controls the shift magnitude. This represents tungsten (warm) versus daylight (cool) lighting and mixed color temperatures.

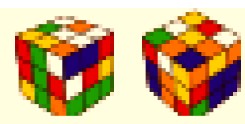

Figure 41: $\delta = 0.12$

**Ambient Color Overlay**  Uniform ambient illumination is modeled by alpha-blending with a constant color: $I' = (1 - \alpha)I + \alpha\,c$. The overlay color $c$ and opacity $\alpha \in [0, 1]$ define the effective fill light. The result approximates global tints arising from wall reflections, skylight, or translucent enclosures.

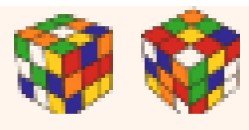

Figure 42: $\alpha = 0.2, \boldsymbol{c} = \#ffd1a4$

**Directional Light with Optional Vignetting**   After gamma-based exposure adjustment, a directional lighting field modulates intensity according to a cosine-like pattern aligned with a unit vector $\boldsymbol{d}$ at angle $\theta$. With normalized coordinates $\hat{\boldsymbol{x}} \in [-1,1]^2$, the mask $m(\boldsymbol{x}) = \mathrm{clip}\left(\frac{1}{2}\hat{\boldsymbol{x}}\cdot\boldsymbol{d} + \frac{1}{2}\right)$ yields $I' = \mathrm{clip}\{I^{1/\gamma} \odot (A + \kappa m)\}$. A radial vignette $v(r) = 1 - \eta r$ optionally attenuates corners. Parameters are $\gamma > 0$, direction $\theta$, directional strength $\kappa \geq 0$, ambient $A \in [0,1]$, and vignette strength $\eta \in [0,1]$. This emulates off-axis window light, desk lamps, and lens vignetting.

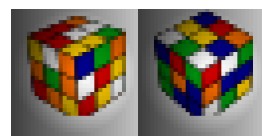

Figure 43: $\gamma = 1, \theta = 0, \kappa = 1, A = 0.5, \eta = 0.5$

**In-Plane Rotation**   A rigid planar transform rotates the image about its center, $\boldsymbol{x}' = \mathbf{R}_\theta(\boldsymbol{x} - \boldsymbol{c}) + \boldsymbol{c}$, with resampling at inverse-warped coordinates to preserve grid spacing. The angle $\theta$ controls orientation. The augmentation models camera roll, tripod tilt, and handheld orientation drift.

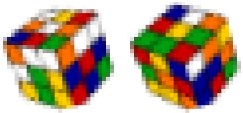

Figure 44: $\theta = 15$

**Isotropic Zoom**   Uniform scaling by factor $f$ about the image center modifies field of view: $f > 1$ crops the center region and upsamples. $f < 1$ downsamples and pads to original size. The single parameter $f > 0$ governs magnification. This simulates focal changes, reframing, and subject distance variation.

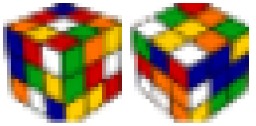

Figure 45: f=1.2

**Edge Cropping with Rescaling**   A fraction $\kappa$ of one edge is removed and the remaining content is resized back to the original dimensions, conserving aspect ratio. Parameters are the crop fraction $\kappa \in (0, 0.5)$ and the selected edge. The effect models partial occlusions, misframing, and sensor readout loss.

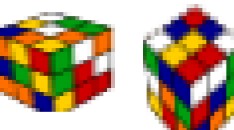

Figure 46: $\kappa = 0.2$

**Additive Gaussian Noise** Zero-mean i.i.d. noise is added per pixel, $I' = \text{clip}(I + \varepsilon)$, $\varepsilon \sim \mathcal{N}(0, \sigma^2)$. The standard deviation $\sigma > 0$ controls grain strength. This simulates sensor read noise and compression residue at moderate bitrates.

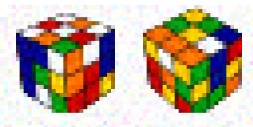

Figure 47: $\sigma = 0.05$

**Salt-and-Pepper Noise** Impulsive corruption sends a proportion $p$ of pixels to intensity extremes, with salt ratio $\rho$ governing the fraction set to white versus black. Parameters are the amount $p \in (0, 1)$ and salt ratio $\rho \in [0, 1]$. This reproduces stuck pixels, transmission dropouts, and packet loss artifacts.

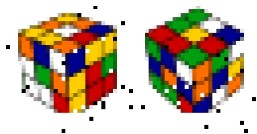

Figure 48: $p = 0.01, \rho = 0.5$

**Poisson (Shot) Noise** Photon-counting statistics are modeled by $I' = \frac{1}{K} \text{Poisson}(KI)$, where $K$ is an exposure scale. As $K$ decreases, variance grows relative to mean, especially in dark regions. This augmentation captures low-light and short exposure regimes.

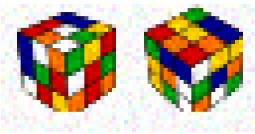

Figure 49: K=256

**Multiplicative Speckle Noise** Granular fluctuations modulate intensity multiplicatively: $I' = \text{clip}(I \odot (1 + n))$ with $n \sim \mathcal{N}(0, \sigma^2)$. The variance $\sigma^2$ shapes speckle contrast. The effect characterizes coherent imaging artifacts (e.g., ultrasound, radar) and rough surface scatter.

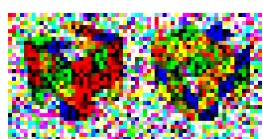

Figure 50: $n \sim \mathcal{N}(0, 1)$

**Color Quantization (Banding)**  Uniform scalar quantization reduces the number of displayable intensities to $L$ levels, $Q_L(I) = \frac{\lfloor LI \rfloor}{L}$. The integer $L \in [2, 32]$ sets the severity. This simulates reduced bit-depth, posterization, and banding from aggressive compression or low-quality displays.

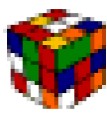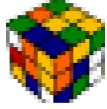

Figure 51: L=8

**Gaussian Defocus Blur**  Convolution with an isotropic Gaussian kernel $G_\sigma$ yields $I' = G_\sigma * I$, approximating defocus from finite aperture. Separability provides efficient realization. The standard deviation $\sigma > 0$ determines blur radius. The augmentation models out-of-focus capture and depth-of-field limits.

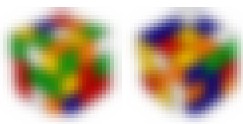

Figure 52: $\sigma = 1.5$

**Horizontal Motion Blur**  Apparent lateral motion is approximated by 1D convolution along the horizontal axis with a box kernel of width $k$, effectively averaging shifted replicas: $I' = b_k *_x I$. The kernel size $k$ (odd) increases streak length. This simulates camera pan or fast object motion.

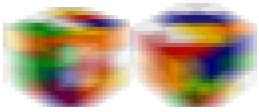

Figure 53: k=9

**Radial (Spin) Blur**  Small-angle rotations around the image center are averaged: $I' = \frac{1}{N} \sum_{i=1}^{N} \mathcal{R}_{\theta_i}(I)$, $\theta_i \in [-\theta_{\max}, \theta_{\max}]$. Parameters are the number of rotations $N$ and maximum angle $\theta_{\max}$. The augmentation represents rotational shake and platform yaw jitter.

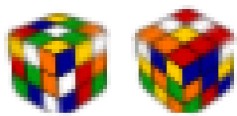

Figure 54: $N = 7, \theta_{\max} = 5$

**Radial Lens Distortion**  For normalized coordinates $\boldsymbol{u}$, a radial warp $\boldsymbol{u}' = \boldsymbol{u}\left(1 + k\|\boldsymbol{u}\|^2\right)$ models first-order barrel ($k < 0$) or pincushion ($k > 0$) distortion. Inverse mapping with interpolation produces the output grid. The distortion strength $k$ controls severity. This reproduces wide-angle lens distortion and optical calibration error.

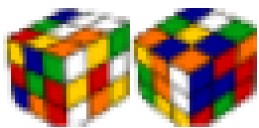

Figure 55: k=0.0001

**Projective (Perspective) Transform**  A homography $\mathbf{H} \in \mathbb{R}^{3\times3}$ maps homogeneous coordinates via $\tilde{x}' \sim \mathbf{H}\tilde{x}$. Choosing $\mathbf{H}$ from perturbed corner correspondences induces keystone deformation. Inverse warping with interpolation preserves sampling density. Parameters are corner displacements or direct specification of $\mathbf{H}$. This models oblique viewpoints and camera pose changes relative to the image plane.

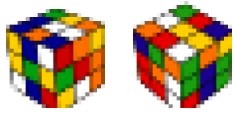

Figure 56: $\Delta = 0.1$

**Object Albedo Channel Offset**  Object face colors are shifted additively in RGB, $\boldsymbol{a}' = \mathrm{clip}(\boldsymbol{a} + \boldsymbol{\delta})$, applied to each semantic region prior to shading. The channel-wise offsets control tint magnitude and direction. This emulates illumination casts and sensor biases producing uniform tints on object surfaces.

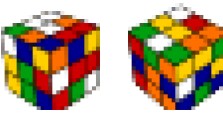

Figure 57: $\boldsymbol{\delta} = [0.1, 0, 0]$

**Object Albedo Channel Scaling**  A diagonal color calibration $\mathbf{D} = \mathrm{diag}(\lambda_R, \lambda_G, \lambda_B)$ scales albedo channels, $\boldsymbol{a}' = \mathrm{clip}(\mathbf{D}\boldsymbol{a})$. Per-channel gains $\lambda$ modulate relative color balance. The augmentation simulates white-balance and per-channel gain mismatches across cameras.

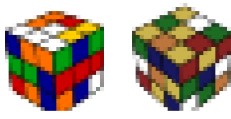

Figure 58: $\lambda_R = 1, \lambda_G = 1, \lambda_B = 1$

**Material Fading (Sticker Wear)**  Chromatic content blends toward local luminance $y = \frac{1}{3}(a_R + a_G + a_B)$: $\boldsymbol{a}' = (1 - \rho)\boldsymbol{a} + \rho\, y\, \mathbf{1}$. The fade factor $\rho \in [0, 1]$ governs desaturation. The effect captures sun bleaching, aging, and pigment loss from surface wear.

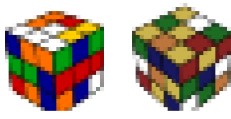

Figure 59: $\rho = 0.5$

**Micro-Texture Perturbations**   High-frequency albedo perturbations are injected as bounded noise $\epsilon \in [-\nu, \nu]^3$ added to each surface region, followed by clipping. The amplitude $\nu$ shapes scratch visibility. This simulates fine scratches and scuffs that locally modulate reflectance.

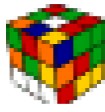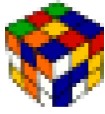

Figure 60: $\nu = 0.05$

**Polygonal Imperfections in Stickers**   Vertices of sticker polygons receive bounded spatial jitter $\Delta v$ with $\|\Delta v\| \le \epsilon$, preserving polygon closure. The piecewise-linear boundary undergoes small shape changes. The jitter radius $\epsilon$ sets deviation magnitude.

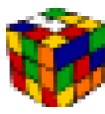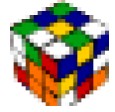

Figure 61: $\epsilon = 0.05$

**Camera Pose Perturbation (Yaw-Pitch-Roll)**   The extrinsic rotation is varied by Euler angles $(\psi, \phi, \varphi)$, and the pinhole projection maps 3D points $\tilde{X}$ to pixels via $\tilde{x} \sim \mathbf{K}\left[\mathbf{R}(\psi, \phi, \varphi) \mid t\right]\tilde{X}$. Small perturbations in the angles capture realistic viewpoint drift while maintaining object visibility. This simulates handheld motion and mount misalignment.

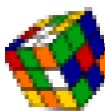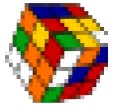

Figure 62: $\psi = -20, \phi = -10, \varphi = -8$

**3D Zoom and Viewport Translation**   Scene geometry is uniformly scaled by $s$, optionally combined with a small translation within the image plane. The projection window adjusts to avoid cropping. Parameters are scale $s > 0$ and planar offsets. The effect simulates effective focal changes and subject recentering without altering the background.

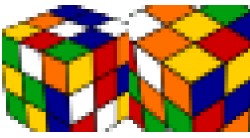

Figure 63: $s = 1.2, offset_x = 0, offset_y = 0$

For Rubik's Cube, we combine these augmentations into the following variants:

- Directional lighting with controllable ambient, shadow, gradient, and subsurface components background image composited behind the cube, camera yaw, pitch, and roll.
- HSV hue, saturation, and value shifts and RGB channel offsets, global contrast and brightness scaling, zoom-in and zoom-out, and positional offsets, camera rotation, and background image.

- A lightweight mix for regularization involving directional lighting, zoom and vertical off-set, camera angle variation, background image, and ambient light tint to simulate global illumination.

- Warm look with glossy material finish for the cube, slight sticker-vertex jitter, per-channel color calibration, warm ambient light and color-temperature shift, zoom and camera jitter, and a background image.

- Harsher directional light with optional vignette, color-temperature shifts in both directions, Gaussian blur and exposure gamma variation, background randomization from the image pool, camera jitter, and low to moderate additive Gaussian noise.

- High-key lighting with white ambient fill, slightly increased contrast and brightness, glossy finish, per-channel scaling, and Gaussian noise.

- Noisy surveillance look with higher Gaussian noise, salt-and-pepper noise, reduced color bit-depth (banding), edge cropping, conservative contrast tweaks, mixed matte or glossy finish, and sticker jitter.

- Controlled illumination using a white ambient fill plus directional light, glossy material, sticker jitter, low noise, slightly boosted contrast, and a background image.

- Camera angle variation, a background image, Gaussian noise to mimic sensor grain.

- Background image, camera jitter, Gaussian blur, directional light with gamma and vignette controls, and light Gaussian noise.

- Background image, directional light combined with a colored ambient wash, motion blur, and Gaussian noise.

- Similar to the above but with lighter ambient, motion blur, a milder vignette, and small Gaussian noise.

- Filled background image, Gaussian blur, radial motion blur, colored ambient light at low opacity, small Gaussian noise, and directional light with vignette.

- Procedural background texture patterns (grid, dots, stripes, crosshatch, waves, or noise) added in the background, Gaussian blur and radial blur, Poisson noise, and directional light on the cube.

- Random opaque shapes placed in front of the scene to partially occlude the cube, combined with Gaussian noise, directional light, camera jitter, zoom, and a background image.

- A simple composition of a background image, directional lighting on the cube, and camera pose variation.

# E  PLANNING RESULT TABLES

Table 2: SPAR's performance on Rubik's Cube across augmentation variants, showing solution length, percentage optimal, nodes generated, time (seconds), nodes per second, and percentage solved.

| DOMAIN | OBSERVATION | LEN | NODES | SECS | NODES/SEC | SOLVED |
|--------|-------------|-----|-------|------|-----------|--------|
| | Bright Lighting | 23.60 | 1.90E+05 | 5.30 | 3.58E+04 | 95.3% |
| | Camera and Background and Ambient Light | 23.66 | 1.91E+05 | 5.30 | 3.60E+04 | 77.4% |
| | Camera and Background and Lighting | 23.62 | 1.91E+05 | 6.31 | 3.03E+04 | 89% |
| | Camera and Background Image | 23.59 | 1.90E+05 | 5.30 | 3.58E+04 | 96.7% |
| | Camera and Background Image | 23.61 | 1.91E+05 | 6.82 | 2.80E+04 | 96.6% |
| | Camera and Lighting | 23.61 | 1.91E+05 | 5.05 | 3.78E+04 | 96.7% |
| | Color Variations | 23.60 | 1.90E+05 | 6.06 | 3.14E+04 | 92.6% |
| Rubik's Cube | Indoor and Glossy | 23.64 | 1.91E+05 | 5.81 | 3.29E+04 | 96.5% |
| | Lighting and Radial Blur and Camera | 23.64 | 1.91E+05 | 6.31 | 3.03E+04 | 88.9% |
| | Lighting and Background and Camera | 23.64 | 1.92E+05 | 5.81 | 3.30E+04 | 79.3% |
| | Minimal | 23.61 | 1.91E+05 | 7.07 | 2.70E+04 | 97.6% |
| | Noisy Camera | 23.61 | 1.91E+05 | 6.06 | 3.15E+04 | 89.8% |
| | Occluders | 23.74 | 1.93E+05 | 6.06 | 3.18E+04 | 78.3% |
| | Outdoor Lighting | 23.75 | 1.93E+05 | 6.82 | 2.83E+04 | 71.1% |
| | Studio | 23.58 | 1.91E+05 | 7.58 | 2.52E+04 | 75.9% |
| | Texture and Poisson Noise | 23.62 | 1.91E+05 | 7.83 | 2.44E+04 | 94.1% |

Table 3: SPAR's performance on Sokoban across augmentation variants, showing solution length, percentage optimal, nodes generated, time (seconds), nodes per second, and percentage solved.

| DOMAIN | OBSERVATION | LEN | NODES | SECS | NODES/SEC | SOLVED |
|--------|-------------|-----|-------|------|-----------|--------|
| | Adversarial Noise | 31.61 | 3.15E+03 | 2.52 | 1.25E+03 | 99.90% |
| | Defocus Blur | 33.12 | 3.30E+03 | 2.64 | 1.25E+03 | 100.00% |
| | Fog Haze | 33.12 | 3.30E+03 | 2.64 | 1.25E+03 | 100.00% |
| | Gaussian Blur | 33.12 | 3.30E+03 | 2.64 | 1.25E+03 | 100.00% |
| | Gaussian Noise | 31.56 | 3.14E+03 | 2.52 | 1.25E+03 | 96.40% |
| | HSV Shift | 33.12 | 3.30E+03 | 2.64 | 1.25E+03 | 100.00% |
| | Indoor Lighting | 31.64 | 3.15E+03 | 2.52 | 1.25E+03 | 99.70% |
| | Lighting Only | 31.58 | 3.15E+03 | 2.52 | 1.25E+03 | 97.70% |
| | Microscopy | 31.65 | 3.15E+03 | 2.52 | 1.25E+03 | 98.50% |
| Sokoban | Motion Conditions | 31.45 | 3.13E+03 | 2.51 | 1.25E+03 | 93.20% |
| | Noise and Distortion | 31.67 | 3.16E+03 | 2.52 | 1.25E+03 | 97.40% |
| | Outdoor | 31.59 | 3.15E+03 | 2.52 | 1.25E+03 | 99.60% |
| | Radial Blur | 31.67 | 3.16E+03 | 2.52 | 1.25E+03 | 99.90% |
| | Rain Effect | 31.68 | 3.16E+03 | 2.53 | 1.25E+03 | 99.90% |
| | Random Occluders | 31.25 | 3.11E+03 | 2.49 | 1.25E+03 | 52.70% |
| | Rotate Image | 31.69 | 3.16E+03 | 2.53 | 1.25E+03 | 99.70% |
| | Surveillance | 31.59 | 3.15E+03 | 2.52 | 1.25E+03 | 98.20% |
| | Zoom Effect | 33.12 | 3.30E+03 | 2.64 | 1.25E+03 | 100.00% |

## F   UNSOLVED EXAMPLES

In this section we include some examples of the test instances that were not solved during planning. These examples show the significant changes in the observations during augmentation.

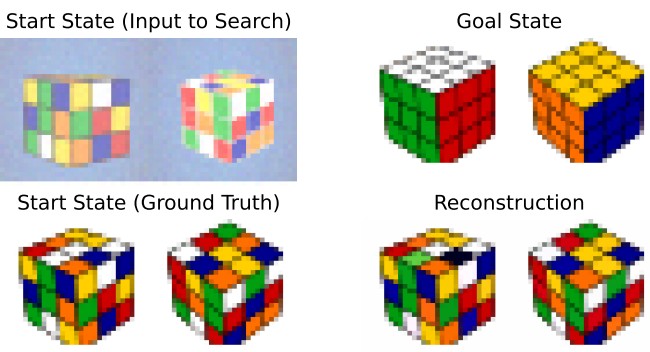

MSE: 0.00254 • Bitwise Equality: 99.00%

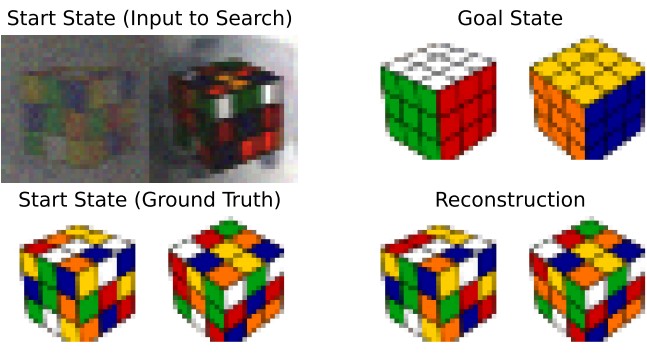

MSE: 0.000584 • Bitwise Equality: 99.25%

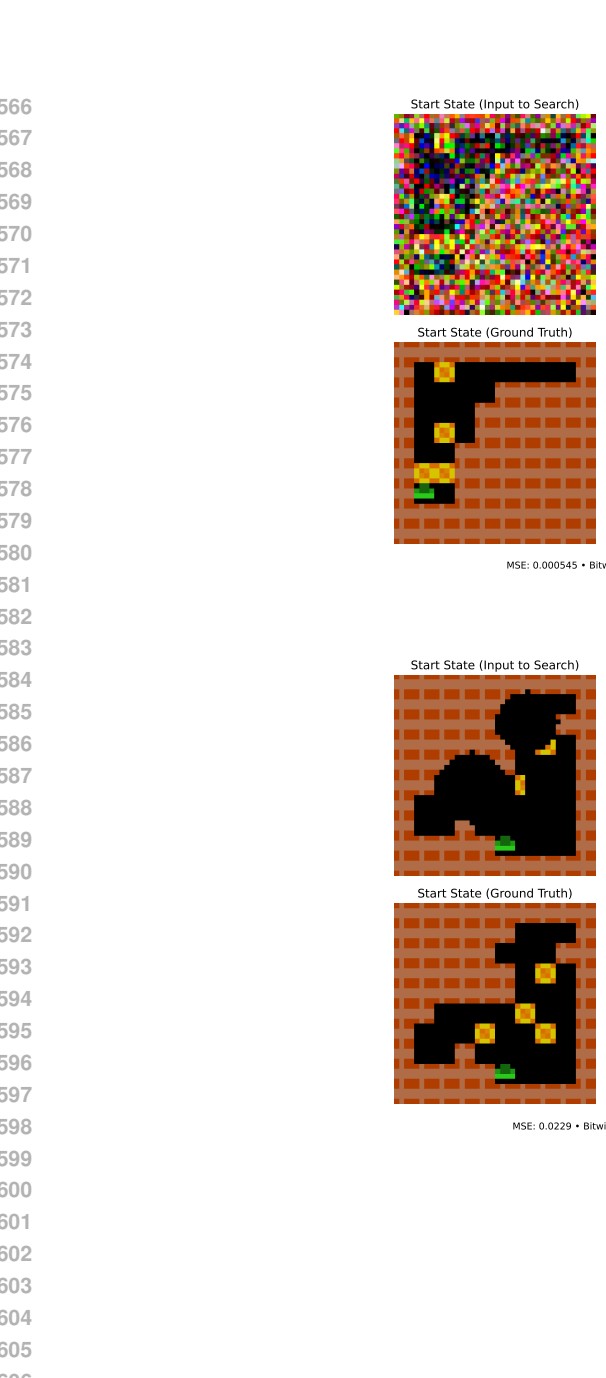

