# OpenReview forum: "Stable Planning through Aligned Representations in Model-Based Reinforcement Learning"
_ICLR.cc/2026/Conference — ICLR 2026 Conference Desk Rejected Submission_

### Official Review · Reviewer_MDoU · 2025-10-30

**Soundness:** 3
**Presentation:** 3
**Contribution:** 3
**Rating:** 6
**Confidence:** 2

**Summary:**

This paper proposes SPAR (Stable Planning through Aligned Representations) — a framework that improves robustness of model-based reinforcement learning (MBRL) under visual or environmental variations. SPAR first trains a discrete world model and a goal-conditioned heuristic function in a clean environment, then introduces a lightweight alignment network that maps visually transformed or noisy observations into the discrete latent space of the world model. This allows the system to perform long-horizon heuristic search without retraining the dynamics or heuristic components when visual changes occur.

**Strengths:**

SPAR demonstrates strong performance on Rubik’s Cube and Sokoban domains, achieving >89% success under 17 different visual transformations and even transferring to real-world Rubik’s Cube images, all while reducing retraining time by ≥95% compared to retraining the full world model or heuristic.

**Weaknesses:**

1. The alignment model requires paired variant–base observations, which may be unrealistic in some real-world settings without synchronized data collection.
2. SPAR’s planning framework is not yet extended to continuous-control domains.
3. The results primarily report quantitative success rates and MSE; visualizations of latent alignment quality or semantic consistency are limited.

**Questions:**

More discussions about Weaknesses.

---

> ### Author Response · Authors · 2025-11-30
>
> We thank the reviewer for the constructive feedback and for the positive assessment. Below we address the concerns raised by the reviewer.
>
> > 1. The alignment model requires paired variant–base observations, which may be unrealistic in some real-world settings without synchronized data collection.
>
> Our detailed discussion of the paired data assumption and its practical implications is provided in the global reponse. For this comment we refer the reviewer to the global response.
>
> ---
>
> > 2. SPAR’s planning framework is not yet extended to continuous-control domains.
>
> The limitations and possible extensions of SPAR to continuous control settings are addressed in the global response. For this comment we refer the reviewer to the global response.
>
> ---
>
> > 3. The results primarily report quantitative success rates and MSE; visualizations of latent alignment quality or semantic consistency are limited.
>
> We appreciate this suggestion and agree that additional qualitative analysis improves clarity. In the revised version we have added more visualizations of aligned observations and their reconstructions.

---

### Official Review · Reviewer_hqcv · 2025-10-31

**Soundness:** 2
**Presentation:** 2
**Contribution:** 2
**Rating:** 4
**Confidence:** 3

**Summary:**

The paper introduces SPAR: train a discrete world model + goal-conditioned heuristic on a clean MDP; then learn a lightweight alignment network that maps transformed observations into the same binary latent so planning can use bitwise equality with rounding. Claims: solves >90% of Rubik’s Cube instances under 17 visual transformations and handles real photos; adaptation requires training only the alignment network and “reduces re-training time by at least 95%.” SPAR uses Q* search with exact latent goal tests; planning results are reported in Table 1 (clean/augmented/real).

Key empirical findings: Rubik’s augmented success 89.39%; real-world success 50%; DeepCubeAI and greedy 0% off-domain; )  Stability: over 10k steps, discrete latents remain accurate while continuous drift, across 17 perturbations.  The alignment model is trained with paired (variant, base) frames at the same timestep, minimizing MSE to the rounded discrete codes; inference rounds A(s̄)

**Strengths:**

Simple, modular adapter: keep (E,T,D, heuristic) fixed; adapt perception via A + rounding bottleneck for exact state re-ID during search.

Long-horizon stability evidence: discrete vs continuous 10k-step rollouts + MSE curves convincingly show drift in continuous latents and stability in discrete.

Real-image demo: proof-of-concept transfer to photos (50% success) beyond synthetic perturbations.

Simple and Elegant Solution: The core idea of decoupling the dynamics/planning module from the perception module is intuitive. Using a lightweight, separate alignment network to map new observations to a fixed, stable latent space is a clean and simple approach.

**Weaknesses:**

Limited baselines: The paper does not compare against natural alternatives such as: training the encoder/world model with standard strong augmentations/domain randomization and then planning (how far does that go?),

Paired data assumption: Training the alignment network requires paired variant/base frames at the same timestep. That’s realistic in sim, but non-trivial in many real settings—precisely where adaptation matters most. The paper acknowledges the need for pairs but does not explore unpaired/weakly-paired regimes or data-efficiency curves.

Theory: This is primarily empirical/engineering; there are no formal guarantees (e.g., on rounding margins, alignment error bounds vs search completeness). That’s fine in principle, but then the empirical section should be more comprehensive on ablations/baselines.

Vague statement and lack citations: "Although, deep learning techniques have shown promise in this area, they need to meet certain requirements to be usable for learning a heuristic function.". Please cite the techniques. Also please explain the certain requirements.

Unclear definition of the “noise” and the “noise transformation module” (Figure 1) in the introduction: The intro refers to Figure 1 but never formalizes the observation domains or the transformation. It’s ambiguous whether “noise” is (i) photometric (blur, brightness), (ii) geometric (crop, rotate), (iii) compositional (occlusion, background), or all of the above; whether transforms are deterministic or sampled from a distribution; whether compositions of transforms occur; and how this “noise” operator interfaces with the alignment module. It would be better to give explanation in the introduction.

**Questions:**

Baselines: How does SPAR compare against training the encoder/world model with strong visual augmentations and then planning (i.e., no alignment adapter)? Please include success rates and compute. (This is the most important missing baseline.)

On the Paired Data Assumption: Could you clarify how you see this method being applied in a real-world setting where paired "clean" states $s$ are not available for the "noisy" states $\overline{s}$? Does this not limit the method to sim-to-sim adaptation, or do you have a path toward unsupervised alignment?

On Failure Cases: The method fails on ~11% of augmented data and 50% of real-world data. This implies the method is extremely brittle to small errors. Could you comment on this sensitivity? Is the planner's success entirely dependent on the alignment network achieving 100% bitwise accuracy?

On Equation 2 and Figure 1: Could you please provide the correct, complete definition for the transition loss in Equation 2? e.g. detach() Furthermore, could you provide a revised diagram that more clearly illustrates the data flow for both training the alignment network and for performing planning at inference time?

---

> ### Author Response · Authors · 2025-11-30
>
> We thank the reviewer for the thoughtful and detailed feedback. Below we respond to the concern raised by the reviewer.
>
> ## Theory and guarantees
>
> Our method relies on a discrete latent space in which rounding induces a margin of safety. As long as the alignment network maps noisy observations to latents that round to the correct discrete codes, the planner's performance is unchanged. If the aligned latent stays within half a quantization bin of the true discrete code, rounding recovers the correct code, and since the planner operates only on discrete codes, residual errors within this margin do not affect planning.
>
> For the discrete world model and encoder, if the learned discrete latent representation is meaningful, predictable, and approximately Markovian, then we can roll it out for many steps while directly checking how many bits match. This enables reliable state comparison, state re-identification, and goal testing. In contrast, continuous latents do not permit exact equality checks, state comparison is less direct, and small errors accumulate over time, which degrades planning performance.
>
> ## Missing baseline: augmentations-only
>
> SPAR is designed as a post hoc adaptation method that reuses an existing world model and planner without retraining, in contrast to training the world model and heuristic from scratch with augmentations. To directly address this concern, we trained a discrete world model on the same dataset as SPAR, using SPAR's alignment model architecture for the encoder and the same hyperparameters. In this setting, the discrete world model predicts next discrete latent states with roughly 50% accuracy on the test set.
>
> On the other hand, when we train the encoder, which is smaller and architecturally simpler than the alignment model, on clean observations only, it easily reaches 100% accuracy on clean test observations. We then train the alignment model to predict only the discrete latent bits of this frozen encoder, using supervised learning on noisy observations, and obtain approximately 90% accuracy on the noisy test set. This comparison shows that SPAR's adaptation approach is more effective than training the world model from scratch with augmentations on the same data.
>
> ## Paired data in real-world settings
>
> Please see the global response for a detailed discussion of the paired data assumption.
>
> ## Failure cases & sensitivity
>
> The reported rates, roughly 11% failure on augmented states and 50% on real photographs, arise in deliberately challenging out of distribution settings rather than from small pixel level perturbations. In our augmented and real world datasets we apply strong geometric and photometric corruptions such as large viewpoint changes, random crops and translations, and substantial noise, which move observations far from the training distribution. In practice, failure cases correspond to heavily occluded or distorted states in which key stickers are barely visible, a regime where even state of the art vision models are known to degrade under distribution shift.
>
> Prior work shows that standard classifiers can assign high confidence to out of distribution examples even when accuracy collapses [1]. Other studies demonstrate substantial degradation in accuracy and calibration under realistic dataset shifts even for methods designed for robust uncertainty estimation or distributionally robust optimization [2][3]. Recent surveys emphasize that robust behavior outside the training distribution remains an open challenge and that evaluation under shift is itself an active research topic [4][5]. Our real world evaluation should be interpreted in this context and is similar in spirit to domain randomization setups that deliberately introduce a large reality gap between train and test [6][7].
>
> Conceptually, SPAR does not require the alignment model to achieve 100\% bitwise accuracy on the entire input distribution. The discrete planner only needs the latent representation to be correct on the start observation, on the goal observation, and on at least one sequence of states that connects them in latent space. Misalignment on states that are never visited by a successful plan does not affect success on that instance.
>
> Our evaluation protocol is intentionally strict. We declare an instance solved only if search reaches a latent code that matches the goal code bit by bit. Under this criterion, a single flipped bit in either the start or goal code is enough to mark a problem as unsolved, even if the predicted code is semantically very close. The reported success rate is therefore a lower bound on the fraction of instances for which the latent plan is qualitatively correct.
>
> We agree that improving robustness to large out of distribution shifts is important. The SPAR framework is compatible with standard techniques from the robustness and OOD literature, and incorporating such methods is a promising direction for future work.

---

> > ### Author Response · Authors · 2025-11-30
> >
> > ### References
> >
> > [1] Hendrycks, D., & Gimpel, K. (2016). A baseline for detecting misclassified and out-of-distribution examples in neural networks. *arXiv preprint arXiv:1610.02136*.
> >
> > [2] Ovadia, Y., Fertig, E., Ren, J., Nado, Z., Sculley, D., Nowozin, S., Dillon, J., Lakshminarayanan, B., & Snoek, J. (2019). Can you trust your model's uncertainty? Evaluating predictive uncertainty under dataset shift. *Advances in Neural Information Processing Systems, 32*.
> >
> > [3] Sagawa, S., Koh, P. W., Hashimoto, T. B., & Liang, P. (2019). Distributionally robust neural networks for group shifts: On the importance of regularization for worst-case generalization. *arXiv preprint arXiv:1911.08731*.
> >
> > [4] Yang, J., Zhou, K., Li, Y., & Liu, Z. (2024). Generalized out-of-distribution detection: A survey. *International Journal of Computer Vision, 132*(12), 5635-5662.
> >
> > [5] Yu, H., Liu, J., Zhang, X., Wu, J., & Cui, P. (2024). A survey on evaluation of out-of-distribution generalization. *arXiv preprint arXiv:2403.01874*.
> >
> > [6] Tobin, J., Fong, R., Ray, A., Schneider, J., Zaremba, W., & Abbeel, P. (2017). Domain randomization for transferring deep neural networks from simulation to the real world. In *2017 IEEE/RSJ International Conference on Intelligent Robots and Systems (IROS)* (pp. 23-30).
> >
> > [7] Chen, X., Hu, J., Jin, C., Li, L., & Wang, L. (2021). Understanding domain randomization for sim-to-real transfer. *arXiv preprint arXiv:2110.03239*.

---

### Official Review · Reviewer_pAHV · 2025-11-07

**Soundness:** 3
**Presentation:** 2
**Contribution:** 2
**Rating:** 4
**Confidence:** 3

**Summary:**

This paper extends DeepCubeAI [Agostinelli et al., 2025] to handle distorted observations by learning an alignment model that maps augmented images to representations of their ground truth versions. This aims to make model-based search methods applicable to more realistic settings. However, while the motivation is strong, the empirical evaluation is limited and the presentation lacks clarity in several places.

**Strengths:**

1. The paper tackles a relevant and challenging problem: enabling model-based planning methods like DeepCubeAI to operate directly on inputs closer to real-life images.

2. Evaluation on the two included environments is reasonable.

**Weaknesses:**

1. The proposed discrete world model may not generalize to continuous environments such as robotics, limiting its practical scope.

2. The evaluation is limited to only two environments (Rubik’s Cube, Sokoban), with only Sokoban being long-horizon.

3. More diverse tasks (continuous or non-symbolic domains) would strengthen the paper.

4. Training the alignment model requires access to clean, undistorted data. This is not a realistic assumption.

5. Figures 3 and 4 are not well explained; it’s not obvious what new insight they add beyond prior work.

6. Line 54 seems to contain a typo.

7. Lines 44-48 could be rewritten to be clearer.

8. Lack of related work, for instance Model-based visual planning with self-supervised functional distances [Tian et al., ICLR 2021], which seems directly relevant.

9. The symbols in Equation 2 make it harder to read it.

**Questions:**

1. How robust is this approach to unseen augmentations at test time?

2. Should the “Rollout stability and reconstruction” paragraph point to a figure?

3. The latent space is 400 dimensional. How many of those dimensions are actually used by the encoder model?

4. Line 278: “However, it is possible to use a new dataset that includes visual variations and clean images that are not present in the original dataset.” – What is the difference in performance if you do that?

5. How would using a discrete world model work if used in an environments that do not have a discrete state space such as for instance robotics?

---

> ### Author Response · Authors · 2025-11-28
>
> We thank the reviewer for the thorough evaluation and for outlining both the strengths and concerns of the work.
>
> > Scope: discrete vs. continuous
>
> We agree that our current experiments are on inherently discrete puzzles, even though SPAR processes RGB observations. In fact, any continuous model can be made discrete by multiplying size by 32 and using binary representations. Empirically, discrete latent world models learned from images are effective for control in environments with continuous state and action spaces. DreamerV2 uses categorical latent variables and reaches human level performance while also solving continuous control tasks from pixels [1]. PlaNet plans entirely in a compact latent state space and solves contact rich continuous control from images [2]. DreamingV2 reports strong results on simulated 3D robot arm tasks using discrete latent world models trained without reconstruction [3].
>
> These findings align with state abstraction theory. Li, Walsh, and Littman [4] show that clustering ground states into abstract states can preserve optimal or near optimal behavior when transitions and values are approximately preserved within each cluster. From this view, a discrete latent world model learned from continuous observations defines an abstract MDP whose states are equivalence classes of continuous states that are behaviorally similar for decision making.
>
> Robotics often solves continuous problems via discrete abstractions. Probabilistic roadmaps build a finite graph in a continuous configuration space and plan on this graph [5]. Gopalan et al. [6] formalize Abstract MDP that layer a high level MDP over a large or continuous base MDP.
>
> Extending SPAR to continuous domains would follow the same pattern. First learn a discrete latent world model from RGB observations in a continuous control environment, inducing a finite latent state set and a learned transition. SPAR’s alignment model then uses pairs of clean latent states and corrupted observations to map back into this discrete latent space. Under standard abstraction theory  [4], if latent states group continuous states with similar transition and reward structure, planning and search in the abstract MDP remain near optimal for the underlying continuous system. We will therefore position SPAR’s application to robotics and other continuous domains as a natural and well motivated direction for future work.
>
> > Data assumption
>
> Please see the global response for more details on the data requirements for training the alignment model.
>
> > Robustness to unseen augmentations at test time
>
> Robust generalization requires training augmentations that cover a wide range of visual transformations. With sufficiently diverse perturbations, the alignment model learns features that transfer to unseen augmentations. We test SPAR on real camera images that differ significantly from the synthetic augmentations used during training. SPAR aligns these real images to the discrete latent space in 50% of test cases, enabling effective planning.
>
> > Latent dimensionality usage
>
> A standard $3\times3$ Rubik’s Cube has $4.3 \times 10^{19}$ reachable states. The minimum number of binary dimensions needed to uniquely index that many states is $\lceil \log_2\!\left(4.3 \times 10^{19}\right) \rceil \approx 66$. Any encoder that assigns a globally unique binary latent code to every cube state must use at least about 66 independent bits. Our 400 dimensional binary latent space uses $400$ bits of $0$s and $1$s and thus provides slack for a representation that is robust and easy to predict rather than pushing the information theoretic limit.
>
> > Dataset variation
>
> Performance is similar as long as the new dataset covers the same types of visual variations. The key requirement is that the alignment model sees sufficient diversity during training to generalize to the test time augmentations.
>
> ---
>
> ### References
>
> [1] Hafner, D., Lillicrap, T., Norouzi, M., & Ba, J. (2020). Mastering Atari with discrete world models. *arXiv preprint arXiv:2010.02193*.
>
> [2] Hafner, D., Lillicrap, T., Fischer, I., Villegas, R., Ha, D., Lee, H., & Davidson, J. (2019). Learning latent dynamics for planning from pixels. In *International Conference on Machine Learning* (pp. 2555-2565). PMLR.
>
> [3] Okada, M., & Taniguchi, T. (2022). DreamingV2: Reinforcement learning with discrete world models without reconstruction. In *2022 IEEE/RSJ IROS* (pp. 985-991). IEEE.
>
> [4] Li, L., Walsh, T. J., & Littman, M. L. (2006). Towards a unified theory of state abstraction for MDPs. *AI&M*, 1(2), 3.
>
> [5] Kavraki, L. E., Svestka, P., Latombe, J. C., & Overmars, M. H. (2002). Probabilistic roadmaps for path planning in high dimensional configuration spaces. *IEEE Transactions on Robotics and Automation*, 12(4), 566-580.
>
> [6] Gopalan, N., Littman, M., MacGlashan, J., Squire, S., Tellex, S., Winder, J., Wong, L., et al. (2017). Planning with abstract Markov decision processes. In *Proceedings of the ICAPS* (Vol. 27, pp. 480-488).

---

### Official Review · Reviewer_8NTx · 2025-11-09

**Soundness:** 2
**Presentation:** 3
**Contribution:** 2
**Rating:** 2
**Confidence:** 3

**Summary:**

SPAR is a method that learns how to make "pathfinding via learning discrete latent world models and using DQN"(DeepCubeAI) robust. It incrementally builds on the concept of DeepCubeAI that learn a disceret latent world model where every observation is encoded to an exact discrte latent state. SPAR's main contribution is to train an alignment model that maps noisy version of an observation (some jitter, orientation, noisy addition) to the exact latent state of its clean version. Then the alignment model can be used with a large scale DeepCubeAI setup.

**Strengths:**

The alignment model can be trained fast since it is a "smaller model" and can be adapted to large-scale pretrained DeepCubeAI setup, making it decoupled and more robust to noisy observations.

The alignment model helps DeepCubeAI scale to real-world.

**Weaknesses:**

Novelty: The main concern I have with the paper is novelty. Based on my understanding, the only novel idea in this paper is "mapping noisy observations to the same latent state of clean observations via MSE" to reuse the large-model. The paper follows the exact same protocol as DeepCubeAI for all other things, including experimental setup.

Experiment exhaustiveness: The paper shows results only for the rubics cube solving task while the augmentations in fig 2 is shown for both cube and sokoban. The experimental results in table 1 are also pretty obvious that vanilla DeepCubeAI won't work in the presence of augmentations since it was not in training data.

Data requirements: The paper mentions 10000 episodes (each with 30 steps) per augmentation (around 30) to train the alignment model. This challenges the scalability of the method. I believe for a task like rubik's cube, it is very important to understand each state (which can be a lot) and also consider "all" augmentations of each state (and that is the reason behind such a large number of datapoints).

**Questions:**

See weaknesses above.

---

> ### Author Response · Authors · 2025-11-28
>
> We thank the reviewer for the thorough evaluation and for outlining both the strengths and concerns of the work. We address novelty, experimental coverage, and data requirements below.
>
> > **Weaknesses:** Novelty ...
>
> To the best of our knowledge, SPAR is the first framework that preserves accurate long-horizon planning under substantial visual transformations while reusing the original world model and heuristic without retraining. Prior visual planners perform explicit retraining on the new domain or joint learning across domains rather than post-hoc adaptation of an existing planner. SPAR explicitly targets the setting where (i) the underlying MDP is fixed and (ii) observations undergo transformations that should preserve the underlying state.
> This adaptation framework differs from robust world-model approaches that modify the world model itself, such as HRSSM [2], which changes the latent dynamics structure.
> Finally, SPAR enforces a strictly frozen world model and heuristic during adaptation. Only the small alignment network is trained on transformed data. Many robust world-model and domain-adaptation methods instead treat robustness as a property of the representation or dynamics and therefore retrain or modify the world model and encoders. We targets the practical scenario where a new visual domain is encountered after training and retraining the world model is infeasible or undesirable.
>
> > **Weaknesses:** Experiment ...
>
> We updated the experiments to include Sokoban results under visual augmentations. We include DeepCubeAI as a baseline to demonstrate that without adaptation the original world model fails under v`isual shifts. This highlights the need for an adaptation mechanism.
>
> > **Weaknesses:** Data ...
>
> A standard $3\times3$ Rubik's Cube has $43{,}252{,}003{,}274{,}489{,}856{,}000 \approx 4.3\times10^{19}$
> reachable configurations. This is the number of valid cube states that can be reached by legal face turns.
> In contrast, the alignment model sees on the order of $10{,}000 \text{ episodes} \times 30 \text{ steps} \times 30 \text{ augmentations} \approx 9\times10^{6}$ noisy-clean pairs, which is roughly $\frac{9\times10^{6}}{4.3\times10^{19}} \approx 2\times10^{-13}$ of the full state space. We are very far from observing every cube state or every augmentation of every state and no existing learning based Rubik's Cube solver comes close to such coverage. As noted by the reviewer, the state space is huge and a robust solver must generalize far beyond observed states. This is exactly the regime in which prior work on Rubik's Cube and related planning problems operates.
> Large scale sim2real work on Rubik's Cube. OpenAI's Rubik's Cube solver trains both control and vision models entirely in simulation using automatic domain randomization and emphasizes the need for vast training data to obtain robust sim2real transfer. The synthetic experience there is far beyond what we use for alignment. [1].
> Within this landscape, our data budget is modest, and much smaller than the data used in high profile sim2real methods.
>
> Below we include detailed results of SPAR on visual variants of the Rubik's Cube environment, where the alignment model is trained on data from 500 episodes of 15 steps.
>
> | Variant                   | Solved (%) | Avg bit equality (%) |
> | ------------------------- | ---------: | -------------------: |
> | base                      |      91.80 |                99.98 |
> | bright_lighting           |      41.60 |                99.68 |
> | cam_and_light             |      61.20 |                99.64 |
> | camera_and_image          |      59.20 |                99.55 |
> | camera_background_ambient |      17.70 |                99.03 |
> | camera_background_light   |      27.90 |                99.48 |
> | camera_image_background   |      49.40 |                99.75 |
> | color_variations          |      42.40 |                99.62 |
> | indoor_glossy             |      46.40 |                99.72 |
> | light_background_camera   |      21.00 |                99.26 |
> | light_radial_cam          |      24.00 |                99.41 |
> | minimal                   |      51.50 |                99.76 |
> | noisy_camera              |      31.40 |                99.50 |
> | occluders                 |      15.70 |                99.10 |
> | outdoor_lighting          |      17.50 |                97.83 |
> | studio                    |      35.00 |                99.60 |
> | texture_poisson           |      35.50 |                99.58 |
> | **ALL**                   |  **39.36** |            **99.44** |
>
> ---
>
> **References**
>
> [1] Akkaya, I., Andrychowicz, M., Chociej, M., Litwin, M., McGrew, B., Petron, A., *et al.* (2019). *Solving Rubik's Cube with a robot hand*. arXiv preprint arXiv:1910.07113.
>
> [2] Sun, R., Zang, H., Li, X., & Islam, R. (2024). Learning Latent Dynamic Robust Representations for World Models. In Proceedings of the 41st International Conference on Machine Learning (PMLR 235, pp. 47234-47260).

---

### Author Response · Authors · 2025-11-28
**Global Response**

## Global response to common questions raised by reviewers

We thank the reviewers for their careful reading and constructive feedback. Below, we address the main questions and concerns related to our assumptions and scope. We have also updated the paper and incorporated changes based on the reviewers' feedbackthe.

---

## On the paired variant-base assumption

**Why this assumption is reasonable.**

Our alignment model uses pairs of observations that share the same underlying state. This mirrors standard supervised corruption and denoising practice in representation learning. Below, we briefly discuss this assumption.

**Denoising and restoration.**

Denoising autoencoders train on \((\tilde s, s)\) with \(\tilde s\) a stochastic corruption of \(s\). They learn a mapping from corrupted to clean with an L2-type loss and provide robust features for deep architectures [1].
A similar supervised corruption and reconstruction setup underlies deep image restoration where synthetic degradations such as Gaussian noise, blur, and compression artifacts create training pairs [2].

**Paired augmented views in self-supervised learning.**

Modern self-supervised visual representation learning [3] and deep RL from pixels [4] assumes paired augmented views of the same image.

**Sim-to-real via domain randomization.**

Domain randomization trains on randomized renderings of the same simulated scene so that real images look like yet another variation of the simulator. This approximates real diversity without exact pairing for every state [5].

**How SPAR fits.**

In our environments, base observations come from canonical simulator renderings of states and variants are created via controlled augmentations. This follows synthetic pairing used in prior literature and in restoration pipelines. It can be done from offline training data.

### How this extends to real-world settings

We identify regimes beyond pure sim-to-sim adaptation, including cases without recorded clean images.

**Approximate clean views via calibration or synthetic rendering.**

Small sets of canonical or nearly clean views can approximate the planning state.
(i) Short calibration sequences under good lighting and controlled viewpoints.
(ii) Synthetic renders from a simulator used for policy learning.
(iii) Multi-view setups where some cameras are less noisy or more constrained.
These serve as base images while heavier perturbations and real camera conditions produce variants. This mirrors SelfIR, which uses short and long exposure captures as realistic paired supervision [6]. In such settings SPAR's supervised alignment remains applicable without synchronized clean and noisy recordings for every frame.

**Learning pseudo-clean targets from only noisy observations.**

When only noisy observations exist, supervision can be relaxed using self-supervised restoration and Noise2Noise. Mapping one noisy observation to another independently noisy observation of the same underlying signal can approximate training against a clean target under standard noise assumptions [2].

## Does this limit SPAR to sim-to-sim adaptation

Our experiments use simulator-generated variant-base pairs, which is the cleanest setting to validate the idea. The method is not tied to strictly paired clean and noisy images. The literature shows that paired or pseudo-paired views are standard and practical in real systems when simulators, calibration sequences, or multi-exposure captures exist [6]. When exact pairs are unavailable, unsupervised or weakly supervised approaches can align domains at the pixel or latent level without paired supervision [2].

---

## References

[1] Vincent, P., Larochelle, H., Bengio, Y., & Manzagol, P.-A. (2008). Extracting and composing robust features with denoising autoencoders. In *Proceedings of the 25th International Conference on Machine Learning* (pp. 1096-1103).

[2] Lehtinen, J., Munkberg, J., Hasselgren, J., Laine, S., Karras, T., Aittala, M., & Aila, T. (2018). Noise2Noise: Learning image restoration without clean data. *arXiv preprint arXiv:1803.04189*.

[3] Chen, T., Kornblith, S., Norouzi, M., & Hinton, G. (2020). A simple framework for contrastive learning of visual representations. In *International Conference on Machine Learning* (pp. 1597-1607). PMLR.

[4] Laskin, M., Lee, K., Stooke, A., Pinto, L., Abbeel, P., & Srinivas, A. (2020). Reinforcement learning with augmented data. *Advances in Neural Information Processing Systems, 33*, 19884-19895.

[5] Tobin, J., Fong, R., Ray, A., Schneider, J., Zaremba, W., & Abbeel, P. (2017). Domain randomization for transferring deep neural networks from simulation to the real world. In *2017 IEEE/RSJ International Conference on Intelligent Robots and Systems* (pp. 23-30). IEEE.

[6] Zhang, Z., Xu, R., Liu, M., Yan, Z., & Zuo, W. (2022). Self-supervised image restoration with blurry and noisy pairs. *Advances in Neural Information Processing Systems, 35*, 29179-29191.

---

### Note · Program_Chairs · 2026-01-17
**Submission Desk Rejected by Program Chairs**

The following references in this submission do not refer to real documents and/or have major errors in bibliographic information:

 Simone Bagatella, Evangelos Nizou, Alexandre Bizzarri, Tomasz Kornuta, Plinio Moreno, et al. Planning from pixels through graph search. In Advances in Neural Information Processing Systems, 2023.